# Quantum-enhanced nanodiamond rapid test advances early SARS-CoV-2 antigen detection in clinical diagnostics

Alyssa Thomas DeCruz [1,2], Benjamin S. Miller[1,3,4] ✉, Da Huang [1], Max McRobbie[1,2], Felix Donaldson[1], Laura E. McCoy [5], Ciara K. O'Sullivan [6,7], Johannes C. Botha[8,9], Eleni Nastouli [5,8,9] & Rachel A. McKendry [1,2] ✉

Quantum biosensors, which harness quantum effects to detect biomarkers, could address the urgent need for more sensitive rapid diagnostics. Lateral flow tests using nitrogen-vacancy centres in nanodiamond labels offer high sensitivity and robustness by controlling the spin-dependent fluorescence to remove background. This is particularly important in complex and variable clinical samples. However, to date only model systems have been studied with few clinical samples. Here we show results of a clinical evaluation of a spin-enhanced nanodiamond test for SARS-CoV-2 antigen with 103 upper respiratory tract swab samples. We find 95.1% sensitivity (Ct ≤ 30) and 100% specificity benchmarked against RT-qPCR, with no cross-reactivity to influenza A, RSV, and Rhinovirus. Modelling with patient data yields a mean of 2.0-days earlier detection compared to conventional gold-nanoparticle tests (just 0.6 days after RT-qPCR) with 2.2-fold more patients detected on the first day of symptom onset, potentially reducing the transmission risk and protecting populations.

The field of quantum technologies for biomedical applications has experienced remarkable growth in recent years. Quantum sensors, in particular, have emerged as a transformative technology, offering enhancements in sensitivity, spatial and temporal resolution enabled by the interaction of biochemical targets with single or ensemble quantum systems[1]. One of the primary systems being studied for biomedical sensing is the nitrogen-vacancy centre (NV) in diamond[2]. This optically active, defect-bound spin system maintains coherence at room temperature and exhibits spin dependent fluorescence emission, enabling optical readout of the spin state as well as controllable modulation of fluorescence. The NV Hamiltonian is sensitive to a broad range of physical parameters, while the defects can be hosted in bulk diamond or diamond nanoparticles, leading to a broad set of

biomedical sensing applications including; nanoscale nuclear magnetic resonance spectroscopy[3], dynamic particle orientation tracking[4] as well as pH and temperature sensing in cells[5]. Zeeman splitting of the sublevels by external magnetic fields has been utilised in the imaging of cells[6] and individual analytes[7,8]. Fluorescent nanodiamonds (FND) containing NV centres have also been studied for in vivo applications, enabling nanoscale tracking of metabolic processes within living cells[9].

FNDs have also found exciting applications in in-vitro diagnostics such as lateral flow tests (LFTs). Fluorescent labels are commonly used to improve test sensitivity, but often suffer from bleaching, blinking, and are limited by variable background autofluorescence[10,11]. FNDs hold potential to overcome these limitations, owning to their high brightness, photostability, and the ability to selectively modulate the

[1]London Centre for Nanotechnology UCL, London, UK. [2]Division of Medicine, UCL, London, UK. [3]Department of Biochemical Engineering, UCL, London, UK. [4]Department of Medical Physics and Biomedical Engineering, UCL, London, UK. [5]Division of Infection and Immunity, UCL, London, UK. [6]INTERFIBIO Consolidated Research Group, Departament d'Enginyeria Química, Universitat Rovira i Virgili, Tarragona, Spain. [7]Institució Catalana de Recerca Estudis Avancats (ICREA), Barcelona, Spain. [8]Great Ormond Street Institute of Child Health, UCL, London, UK. [9]Advanced Pathogen Diagnostics Unit, UCLH-NHS Trust, London, UK. ✉e-mail: ben.miller@ucl.ac.uk; r.a.mckendry@ucl.ac.uk

signal[12,13]. This is particularly useful in LFTs where the performance of fluorescent reporter particles is often limited by background autofluorescence from the nitrocellulose membrane[14]. Our recent work demonstrated the potential of spin-enhanced FNDs in a rapid LFT, achieving a fundamental detection limit of $8.2 \times 10^{-19}$ M, 94,000-fold more sensitive than conventional gold nanoparticles[15]. In a diagnostic assay, however, this limit is not reached because of nonspecific binding of nanoparticles to the membrane, reflected by the 7500-fold sensitivity improvement achieved for the HIV-1 RNA model assay in the same work[15].

Although the FND platform is broadly target agnostic, Miller et al.[15] largely focused on nucleic acid detection, achieving single-copy amplicon detection using an isothermal amplification step. In direct antigen detection assays, where target amplification is not possible, nanoparticle detection limits translate directly to analytical sensitivities. Consequently, spin-enhanced FND biosensors could be impactful in addressing the need for more sensitive antigen tests for use in resource limited settings such as doctors' surgeries, pharmacies and even self-testing at home. Key challenges for FND-based antigen tests are sensitivity and clinical evaluation. For example, Wei-Wen Hsiao et al.'s[16] limit of detection (LoD) of 1.94 ng/mL for SARS-CoV-2 nucleocapsid protein remains comparable to simpler gold nanoparticle-based LFTs. In later work, Le et al.[17], achieved a LoD of 0.02 ng/mL detecting ESTAT6 for tuberculosis diagnosis but direct antigen detection was limited by the use of cell-cultured samples. In a non-modulated FND approach, Feuerstein et al.[18] demonstrated the feasibility of FND-based LFTs for Ebola glycoprotein detection but encountered sensitivity limitations in serum samples. Notably, sensitivity remains a key challenge and to the best of our knowledge, no FND studies have been reported using direct (non-cultured) clinical samples, with the exception of our proof-of-concept study, Miller et al.[15], where we tested a limited number of clinical samples: one clinical standard plasma sample and a seroconversion panel for the HIV-1 RNA assay. Hence, there remain significant research challenges translating sensitivity in model systems through to clinical samples[18], highlighting the need for large-scale clinical evaluation and clinical benchmarking to gold standard diagnostic methods such as PCR.

The development of antigen LFTs is a key component of the WHO R&D Blueprint strategy to address priority diseases with epidemic potential[19] such as coronaviruses, Crimean-Congo haemorrhagic fever and Ebola where sensitivity is a challenge. Additionally, the Foundation for Innovative New Diagnostics has just published a new Pathogen Diagnostic Readiness Index indicating large unmet needs in diagnostic performance across diseases of epidemic potential (www.finddx.org/data-and-impact/dashboards/diagnostic-readiness-index). In 2019, SARS-CoV-2, a previously unknown virus, emerged infecting over 700 million people globally, leading to >7 million deaths by 2024 (www.data.who.int/dashboards/covid19/cases). SARS-CoV-2 is characterised by its high transmissibility, long incubation period, and high number of asymptomatic carriers[20] - key factors that led to accelerated disease spread and subsequent pandemic. Consequently, timely diagnostics played a key role in the identification of cases to reduce community transmission[21]. Nucleic acid amplification tests (e.g. PCR tests) targeting viral RNA were quickly adopted as the gold standard for COVID-19 diagnosis[22], with antigen-detecting LFTs (Ag-LFTs) later implemented for widespread use at the point-of-care or home settings[23]. Confirmatory testing for COVID-19 has relied on quantitative reverse transcriptase PCR (RT-qPCR) due to its high sensitivity and specificity towards SARS-CoV-2 genes in respiratory tract specimens[24]. The detection limits of RT-qPCR can reach as low as 1-100 copies/mL, useful for early detection in the first 24–48 hrs of infection (Fig. 1a)[25]. Despite the high sensitivity, molecular methods are inherently difficult to scale-up as seen during the COVID-19 pandemic, when RT-qPCR was unable to meet high frequency testing demands due to high cost, personnel, sample transportation and long turn-around times of 1–2 hrs[26]. Ag-LFTs were widely adopted as an alternative to molecular methods to meet the high testing demand[27].

Ag-LFTs target the spike or nucleocapsid viral protein detected from nasal or throat swab samples without the need for additional extraction steps[28]. SARS-CoV-2 Ag-LFTs are low cost, fast (<15 min), and amenable non-traditional healthcare settings[23]. Importantly, antigen positivity correlates well with infectiousness and may therefore hold advantages over RT-qPCR which faces prolonged RNA positivity[29] (Fig. 1a). Yet despite the development of over 1000 commercial antigen tests (www.finddx.org/covid-19/test-directory/), the majority still lack sensitivity, with limits of detection equating to ~$10^5$-$10^6$ copies/mL or Cts<25[30,31]. Furthermore, many LFTs see a significant decline in performance in moderate viral load samples (Ct 25 to 30)[32] and hence positive cases could be missed, with detection occurring up to 1–2 days later than RT-qPCR (Fig. 1a). This window for early detection relies on both high sensitivity and rapid turn-around time, which quantum sensing holds potential to address.

In this work, we present the development of an FND antigen SARS-CoV-2 LFT (Fig. 1b). The analytical sensitivity is evaluated with model recombinant nucleocapsid protein and inactivated whole virus with both Wild-Type and Omicron strains, and specificity is demonstrated against other human respiratory viruses. We undertake a large, blinded clinical evaluation of patient swab samples benchmarked to RT-qPCR, which is then used, with patient data, to model the clinical impact of FND-assay diagnostics.

## Results and discussion

### Spin-enhanced fluorescent nanodiamond characterisation

Antibody-functionalised FNDs are used as the detection nanoparticle forming a sandwich complex between the target analyte and secondary capture antibody (Fig. 1b). Size and morphology of 600 nm FND-PG were observed by SEM (Supplementary Fig. 1).

Figure 1c shows the energy level diagram of an NV⁻ centre with spin triplet ground and optically excited states, and a pair of intermediate metastable singlet dark states. Green light (550 nm) excites electrons into the optically excited manifold, which can then decay back down to the ground state, emitting a photon (~675 nm), where spin is conserved throughout. Alternatively, electrons in the $m_s = \pm 1$ excited state can decay via the singlet 'dark' state, where no visible photon is emitted, in a non-spin-conserving relaxation process. This enables the spin sub-levels to be distinguished via fluorescence intensity - optically detected magnetic resonance (ODMR) - as well as optical initialisation of the population into the $m_s = 0$ state. Once initialised, controllable reduction of the photoluminescence can then be achieved by transferring population from the $m_s = 0$ to the $m_s = \pm 1$ states (increasing the spin population decaying via the dark state) through the application of a driving field with energy resonant with either the ground ($\Delta E = 2.87$ GHz) or optically excited ($\Delta E^* = 1.43$ GHz) zero-field splitting. This reduction in fluorescence is evident in the ODMR spectrum of FNDs immobilised at the test line of a nitrocellulose lateral flow strip in Fig. 1c. Fluorescence is reduced by 0.70% at 1.43 GHz and 2.6% at 2.87 GHz, corresponding to the excited ($\Delta E^*$) and ground state ($\Delta E$) zero field splitting, respectively. Selectively controlling fluorescent intensity provides the ideal mechanism for lock-in detection as the effect is highly specific to the fluorescent source.

Fluorescent intensity scales with number of NV centres per particle[15], therefore larger particles ~600 nm FNDs were selected for this work. In addition, the optical spectrum of NV⁻ centres (Fig. 1d) is well-suited for biosensing applications, away from most biological autofluorescence, which is at shorter wavelengths[33].

### Design of FND-based LFT for SARS-CoV-2 antigen detection

Sourcing high affinity capture reagents is critical to diagnostic sensitivity and specificity, but remains a challenge due to high cost[34,35].

Receptor-ligand binding kinetics and mass transport impact LFT performance, where the ratio of signal from specific analyte-mediated binding to non-specific binding and background signal should be optimised[36]. Poor kinetics limit sensitivity by reducing the fraction of bound antigen, reducing signals. While increasing nanoparticle concentration can increase specific binding rates, it also increases non-specific binding[15].

We identified six antibodies: four commercially available SARS-CoV-2 nucleocapsid antibodies (40143-R001, 40143-R004, 40143-R040, 40143-MM08; Sino Biological) with reported high affinities ($K_D \approx 0.02$ nM), plus two in-house SARS-CoV antibodies (CR3018 and CR3009; UCL, Division of Infection and Immunity). We used Biolayer-interferometry to measure antibody binding affinity ($K_D$) (Fig. 2a) and calculate rate of association ($k_{on}$) and dissociation ($k_{off}$). We found the

$K_D$ values ranged from 0.19 nM to 69 nM and $k_{on}$ values $1.2-3.6 \times 10^4$ $s^{-1}nM^{-1}$. Antibody MM08 (AbMM08) had the highest $K_D$ of 0.19 nM [95% CI: 0.091–0.36] and the fastest $k_{on}$ of $3.6 \times 10^5 s^{-1}M^{-1}$ [95% CI: 3.2–3.9 × 10⁵]. We proceeded with all four commercial antibodies, found to have stronger binding affinities ($K_D$ values 0.19 nM to 0.34 nM), and for their demonstration as high performance antibodies in literature. This included AbMM08, AbR004, and AbR001 ranking amongst the top 5 performing antibody pairs in a large evaluation of 1021 SARS-CoV-2 antibody pairs (Cate et al.[37]) and numerous early reports establishing their high quality during the pandemics initial test development phases[38–40]. Note, antibodies CR3018 and CR3009 were provided early on, previously developed against SARS-CoV[41], whereas the commercial antibodies were later developed specifically against SARS-CoV-2 hence higher binding affinity can also be attributed to

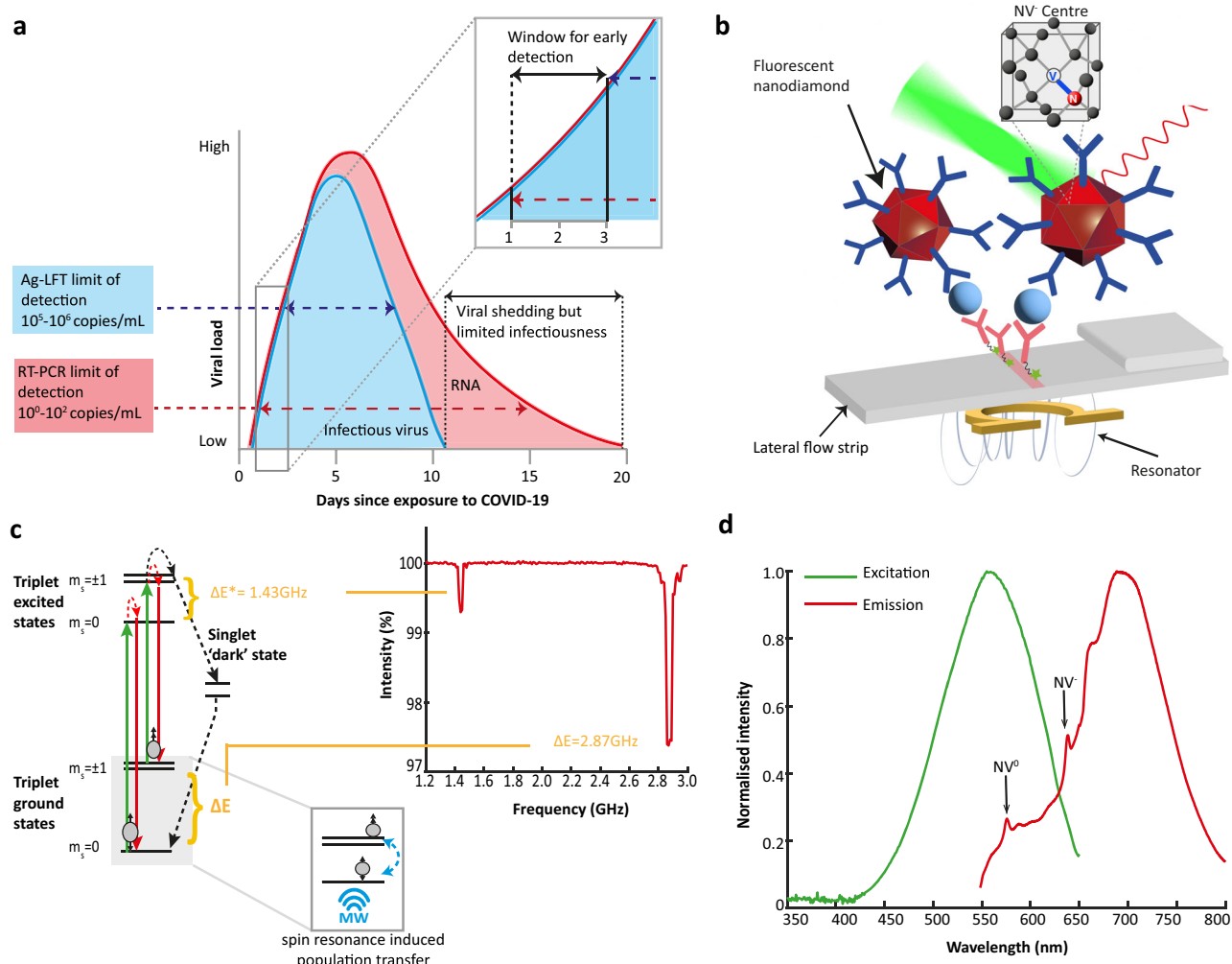

**Fig. 1 | Early detection of SARS-CoV-2 antigen by spin-enhanced LFT and FND characterization. a** SARS-CoV-2 infection dynamics of RNA and infectious virus by viral load with the LoD of RT-qPCR at ~$10^0$-$10^2$ copies/mL, detecting active virus 1–3 days earlier than current antigen-detecting LFTs. Call-out box shows the 1–3 days window for early detection with improved sensitivity of Ag-LFTs. **b** Schematic of FNDs immobilised at the LFT test line via antibody sandwich complex. The FND is excited at 550 nm and emits from the visible to the NIR. A small omega-shaped resonator is driven at the NV⁻ zero-field splitting frequency (2.87 GHz) to induce spin population transfer between the $m_s = 0$ and $m_s = \pm1$ states. The amplitude of this driving field is modulated to provide a time varying fluorescence signal. **c** Energy level diagram of NV⁻ centre, showing the zero-field splitting of the spin triplet optical ground and excited states, the spin conserving optical transitions, and the non-radiative decay pathway from the optically excited

$m_s = \pm1$ state via the metastable singlet states. Under continuous optical illumination, this non-spin-conserving decay pathway results in an increase in population in the $m_s = 0$ state in a process known as optical initialisation. Once a population difference between the $m_s = 0$ or $m_s = \pm1$ is established, if energy resonant with the spin transitions is provided ($\Delta E^* = 1.43$ GHz and $\Delta E = 2.87$ GHz), population is transferred from the $m_s = 0$ to the $m_s = \pm1$ states. Due to the singlet decay pathway not resulting in a visible photon, this reduces the detected photoluminescence. This effect is demonstrated by the results of the continuous wave optically detected magnetic resonance (CW-ODMR) experiment shown in the inset with reductions in fluorescence centred at $\Delta E^*$ and $\Delta E$. **d** NV centre excitation and emission spectra showing an excitation peak at 550 nm and emission peak at 675 nm. Zero phonon lines for NV⁰ and NV⁻ centres are labelled. Source data are provided as Source Data file.

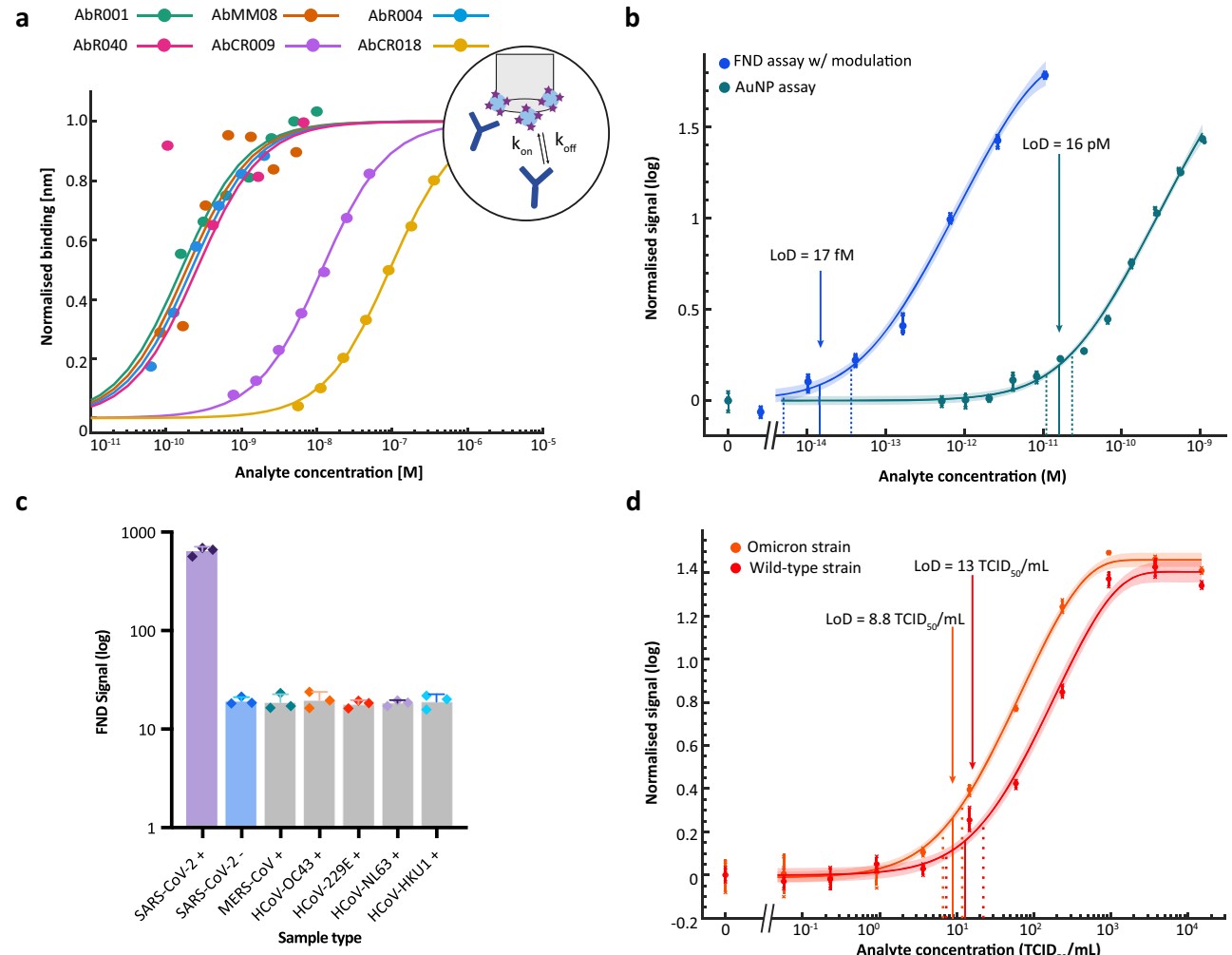

**Fig. 2 | Antibody characterisation and FND Ag-LFT assay analytical sensitivity and specificity. a** Fitted curves of SARS-CoV-2 antibody-antigen binding over a range of antigen concentrations (data points) evaluated using Biolayer-interferometry. Sensor design shown in schematic. **b** Serial dilution of recombinant nucleocapsid protein with FND Ag-LFT evaluated by lock-in modulation analysis (blue) resulting in a LoD of 17 fM [95%CI: 5.9-41] using an exponential fitting method (fit shown as solid line with 95% CI shaded region). Data points show mean ($n = 3$ test replicates) and crosses show individual replicates. This was compared to 40 nm gold nanoparticles (green) with an LoD of 16 pM [95%CI: 11–23]. Data points represent mean ($n = 3$ test replicates) and crosses represent individual replicates. Measurement replicates give a coefficient of variance 73-times smaller than the test replicates coefficient of variance, so measurement variance is negligible, and is excluded from this, and further analysis, but available in the Source Data. LoD shown as solid vertical line with 95% CI (dotted vertical lines) **(c)** Mean FND signal of recombinant antigens from other human coronaviruses, showing no statistically significant difference from SARS-CoV-2 negatives (one-way ANOVA, *p*-value = 0.960, $F = 0.19$, DF = 17, post-hoc Dunnett's multiple comparisons). Bar represents mean signal and SD ($n = 3$ test replicates), data points represent individual replicates. **(d)** Serial dilution of SARS-CoV-2 gamma-irradiated whole virus of both Wild-type (red) and Omicron (orange) strains evaluated by lock-in analysis showing an LoD of 13 TCID$_{50}$/mL [95% CI: 7.3-22] and 8.8 TCID$_{50}$/mL [95% CI: 6.6-12], respectively. Data points represent mean ($n = 3$ test replicates) and crosses represent individual replicates. Exponential fit shown as solid line with 95% CI shaded region. LoD shown as solid vertical line with 95% CI (dotted vertical lines). Source data are provided as Source Data file.

specificity to the target antigen. In our data, antibodies R001 and R040 showed significantly weaker binding than manufacturer reports. Kinetic parameters are detailed in Supplementary Fig. 2 and Supplementay Table 1.

A step-wise approach was used to screen antibody pairs based on optimised binding interactions where first, the binding affinity of the detection antibody was measured, followed by the secondary antibody binding affinity in a sandwich format, minimizing epitope binding interference and non-specific binding. The antibody-functionalised FNDs were first evaluated in direct bind format, targeting biotinylated recombinant nucleocapsid protein. The strong binding affinity of biotin-avidin ensures effective capture at the test line, making the FND-antibody complexation with the analyte the limiting interaction. The signal-to-noise ratio (SNR) is defined as the ratio of the lock-in amplitude of a positive test line compared to the mean lock-in amplitude of

the negative controls, where the signal is due to FNDs non-specifically bound to the membrane, and variation is caused by strip-to-strip variation. Here the SNR of a strong positive sample (10 ng/mL) was used to evaluate antibody-conjugate performance. FND-AbMM08 showed the highest SNR of 84 although there were no significant differences (one-way ANOVA, *p*-value = 0.71, $F = 0.48$, DF = 11, post-hoc Tukey's multiple comparisons), shown in Supplementary Fig. 3a. FND-AbMM08 was selected as the best performing detection antibody for the FND system.

Functionalisation and dispersity of the FND conjugate was measured using dynamic light scattering. The 600 nm FND-PG showed average hydrodynamic diameter at 628 nm (±148) and a poly-dispersity index (PDI) of 24%, consistent with approximate size of particles with a polyglycerol coating (Supplementary Fig. 3b). FNDs functionalised with AbMM08 showed a 20 nm increase in

hydrodynamic diameter to 643 nm (±124) and PDI of 21%, corresponding to the approximate theoretical size of a typical antibody (15 nm). Secondary capture antibody pairs tested with FND-AbMM08 showed that AbR001 out-performed AbR004 (*p*-value = 0.025) and AbR040 (one-way ANOVA, *p*-value = 0.036, $F = 8.2$, DF = 8, post-hoc Tukey's multiple comparisons, Supplementary Fig. 3c). Therefore, FND-AbMM08 and capture AbR001 was used for further assay optimisation and performance evaluation. It is important to note that the antibody pair evaluation was conducted using a consistent recombinant antigen and potential antibody pair selection bias resulting from screening on a single antigen source was not considered in this initial down-selection process[37]. Antibody-printed test lines require high volumes and a large amount of excess antibody therefore we used biotinylated AbR001 (AbR001-b) in solution with polystreptavidin-printed test line to reduce antibody consumption by 84.6% and therefore cost.

The concentration of FND-Ab and capture antibody, and buffer formulations were then optimised. To ensure optimal binding, a high concentration of capture antibody AbR001-b was used to screen through FND-AbMM08 conjugate concentrations ranging from 6.6 fM to 42 pM. The positive and negative signal scaled linearly with increasing FND concentration where the SNR, found by dividing the fitted linear regressions, resulted in a constant value of ~20 (Supplementary Fig. 4a,b). A concentration of 26 fM was selected to minimise reagent consumption while also generating a measurable intensity for the strong positive test line. Buffer components were re-formulated to suit clinical testing of SARS-CoV-2 nasal swabs, which requires viral lysis components in addition to non-specific protein blockers. Final buffer composition found the addition of 0.8% casein and 0.05% Tween20 improved the SNR and IGEPAL CA-630 was found to be an effective detergent for viral lysis (Supplementary Fig. 4c,d).

### Analytical sensitivity of spin-enhanced FND LFT compared with gold nanoparticles

The sensitivity of the FND LFT was first evaluated using a serial dilution of SARS-CoV-2 recombinant nucleocapsid protein where the lock-in fluorescence intensity was used to plot and quantify the analytical LoD using previously reported statistical method applying an exponential curve fit[42]. A comparison of different fitting models is shown in Supplementary Table 2. The assay here presented yielded an LoD of 17 fM [95% CI: 5.9-41] (0.78 pg/mL) with spin-modulated lock-in (Fig. 2b). Note that the signal is also limited by non-specific binding of FNDs to the nitrocellulose - FND signal can still be detected in the negative sample (Supplementary Fig. 6a). To fully exploit the detection sensitivity of spin-enhanced FND detection, non-specific binding should be further reduced below the readout noise floor.

The sensitivity of the FND assay was then benchmarked to that of a conventional AuNP-based assay developed in-house using the same reagents. The AuNP assay yielded an LoD of 16 pM [95% CI: 11–23], demonstrating a ~1000-fold sensitivity improvement with FNDs (Fig. 2b, unpaired two-tailed *t*-test, *p*-value = $8.44 \times 10^{-15}$, T = -14, DF = 29). Test line images are shown in Supplementary Fig. 6b. The analytical sensitivity reached in this work demonstrates superior sensitivity of spin-enhanced LFTs compared to similar low-cost rapid LTFs reported in literature[16,38,43–45] (Supplementary Table 3) and reaches LoDs comparable to high-throughput, microarray platforms such as Quanterix's Simoa. This platform reports a lower limit of quantification (LLOQ) of 0.31 pg/mL and LoD of 0.099 pg/mL [0.046-0.20] but at a substantially higher cost of ~£400,000 for the HDx platform and £2400 per SARS-CoV-2 N protein kit for up to 96 samples. This is compared to an approximate cost of £1.30 per FND strip and a low-cost reusable reader without requiring specialised technologists[15].

The specificity of the FND assay to SARS-CoV-2 was tested against recombinant antigens of other common coronaviruses. No cross-reactivity was observed for MERS-CoV (*p*-value = 0.99), HCoV-OC43

(*p*-value = 0.99), HCoV-299E (*p*-value = 0.95), HCoV-NL63 (*p*-value = 0.99) and HCoV-HKU1 (Fig. 2c, one-way ANOVA, *p*-value = 0.96, $F = 0.19$, DF = 17, post-hoc Dunnett's relative to SARS-CoV-2 negatives). Next, the technical performance was further demonstrated in a more clinically relevant sample, using a serial dilution of the whole inactivated virus samples in VTM. The assay showed the ability to detect both the Wild-Type and Omicron variants with similar sensitivities, yielding LoDs of 13 TCID$_{50}$/mL [95% CI: 7.3-22] and 8.8 TCID$_{50}$/mL [95% CI: 6.6-12], respectively (Fig. 2d, unpaired two-tailed *t*-test, *p*-value = 0.26, $T = 1.2$, DF = 45). This affirms the choice of targeting the highly conserved nucleocapsid protein whereby performance is less likely to be compromised by emerging variants[46]. Furthermore, the viral RNA concentration (ddPCR) of the stock viral isolate enabled the conversion of estimated LoD correlating to ~$3 \times 10^4$ genome copies/mL, which indicates a better LoD than most LFTs on the market with LoDs as high as $10^5$-$10^6$ genome copies/mL[47]. For comparison to a widely used commercially available rapid LFT, FlowFlex assay showed a visible test line around $4 \times 10^3$ TCID$_{50}$/mL using the wild-type viral isolate tested in-house, which is ~13-fold lower than the claimed sensitivity of ~300 TCID$_{50}$/mL and 300-fold less sensitive than the FND assay above (Supplementary Fig. 6c). However, this comparison is less pertinent than the comparison with the in-house gold nanoparticle assay (Fig. 2b), as commercial tests likely use different antibodies, buffers, and membranes, all of which will affect sensitivity, confounding the nanoparticle comparison.

### Blinded study to determine clinical sensitivity of FND Ag-LFT compared to RT-qPCR

The next step was a blind evaluation of the performance of the FND assay using clinical swab samples compared to gold standard RT-qPCR results (Ct value). 103 frozen samples, a combination of nasopharyngeal and nasal swabs, were received from UCLH collected from individuals (returning travellers and symptomatic patients) from Nov-Dec 2022 and Jun-Aug 2023 during the pandemic. The initial sample set was selected at random by UCLH to broadly represent patient sample pool viral loads. A further set of samples was requested to analyse the lower limits of detection i.e. high Ct values. The samples comprised 53 SARS-CoV-2 positive samples with Ct values between 17 and 37, and 50 SARS-CoV-2 negative samples, 13 of which were positive for other respiratory viruses including Flu A, RSV, and RHINO virus. An in-house RT-qPCR standard curve was used to convert Ct values to RNA copies/mL (Supplementary Fig. 7). Sensitivity was determined using ROC analysis with 37 SARS-CoV-2 negative swab samples as the negative control to determine the threshold cut-off. Sensitivity was analysed by grouped Ct values commonly used in literature (Fig. 3a). The FND assay showed positive correlation (correlation coefficient of 0.87 [95% Bayesian credible interval: 0.81, 0.91]) with RNA concentration, detecting samples with concentrations ≥$10^4$ copies/mL (Fig. 3b). The assay maintained 100% sensitivity in high viral load samples (Ct ≤ 25) and 86.8% sensitivity (area under curve: 0.92) across the whole Ct range (17–37) (Fig. 3c, Supplementary Table 4). No significant differences in FND signals or viral loads were found between nasopharyngeal and nasal swabs (unpaired two-tailed *t*-test *p*-values 0.6457 and 0.9886, respectively) as shown in Supplementary Fig. 8. The assay showed high specificity of 100% (n = 50) and found no cross-reactivity in swab samples confirmed SARS-CoV-2 negative and positive for other common respiratory viruses including Flu A (n = 6), RSV (n = 6) and RHINO virus (n = 1). One-way ANOVA with post-hoc Dunnett's multiple comparisons determined no significant differences from the mean of Flu A and RSV samples from SARS-CoV-2 negative controls (all *p*-values = 1.0, $F = 9.37$, DF = 103), and significant differences from the mean SARS-CoV-2 positive (*p*-value = $7.9 \times 10^{-7}$, Fig. 3d). Only one RHINO virus sample was included in this evaluation so statistical significance could not be determined for this category.

Importantly, this sensitivity exceeds the WHO target product profile criteria for SARS-CoV-2 RDTs with acceptable sensitivity of

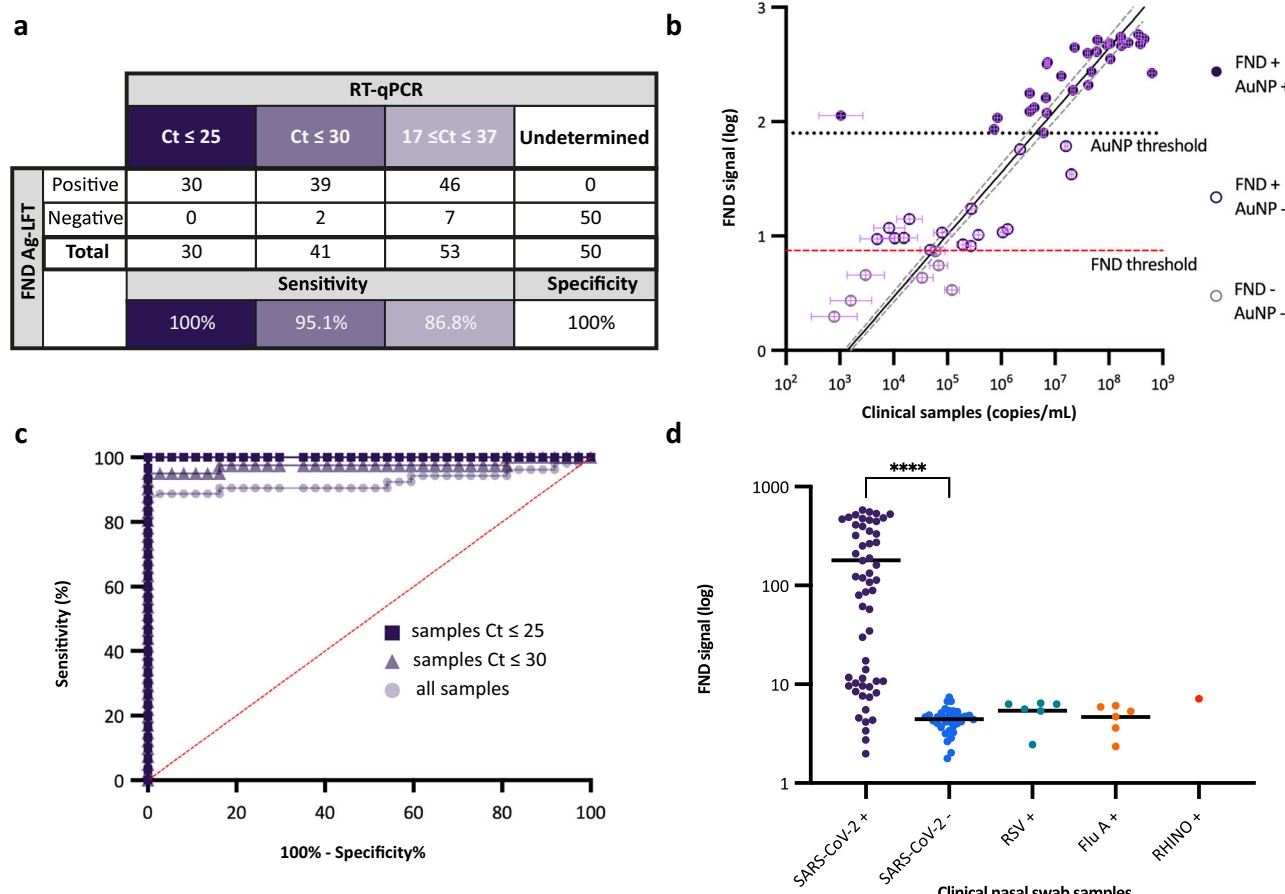

**Fig. 3 | Clinical evaluation of FND Ag-LFT assay. a** Sensitivity and specificity evaluation of FND Ag-LFT compared to RT-qPCR grouped by sample Ct values. Sensitivity = 100%, 95.1%, and 86.8% for Ct ≤ 25, Ct ≤ 30, and all samples, respectively. Specificity was 100% (**b**) Plot of 53 SARS-CoV-2 positive clinical nasal swab samples with FND-Ag test line signal plotted against clinical sample concentrations (copies/mL) determined by RT-qPCR (*n* = 3 test replicates, Supplementary Fig. 8). Circles with error bars show the means and standard deviations of copy number and FND signal for each clinical sample. The data was fitted to a symmetrical Bayesian linear regression with the errors-in-variables in x and y[58] (solid black line with 95% credible interval as grey dashed lines) to account for uncertainty in copies/mL quantification from RT-qPCR and FND signal (Supplementary Fig. 9). Purple indicates FND true positive, purple outline indicates FND positive and AuNP negative, and grey outline indicates RT-qPCR positive and negative on the FND-Ag assay and AuNP assay. Red dashed line indicates FND Ag-LFT cut-off threshold set

to give false positive and false negative rates of 5% using *n* = 37 negative clinical samples (biological replicates). The black dotted line indicates extrapolated threshold for AuNP assay. **c** ROC analysis of clinical swab samples using 37 SARS-CoV-2 negative control samples and 53 positive samples showing sensitivity and specificity trade-off across all samples and sub-categories of samples with Ct ≤ 25 and Ct ≤ 30. **d** Raw data plot of clinical nasal swab samples positive for other respiratory viruses. Black bar represents mean signal. Data points represent distinct clinical samples where *n* = 53 SARS-CoV-2 positive samples, *n* = 50 SARS-CoV-2 negative samples, *n* = 6 RSV positive samples, *n* = 6 Flu A positive samples, and *n* = 1 Rhino Virus positive sample (all biological replicates). One-way ANOVA and Dunnett's post-hoc test was used to determine significance from SARS-CoV-2 negative samples, **** denotes *p*-value = 7.89 × 10⁻⁷ for SARS-CoV-2 positives, *p*-value = 1.00 for RSV + , FluA + , and RHINO virus + ; $F$ = 9.367, DF = 103. Source data and statistical analysis provided as Source Data file.

≥80% for Ct ≈ 25 or 10⁶ genome copies/mL and desirable sensitivity ≥ 90% for Ct ≈ 30 or 10⁴ genome copies/mL, with 100% sensitivity in Cts ≤ 25 and 95.1% in Cts ≤ 30 (Fig. 3c). Many commercially available Ag-RDTs, including FlowFlex SARS-CoV-2 Ag (ACON Biotech) and BD Veritor system for rapid detection of SARS-CoV-2 (Becton Dickinson), lack sensitivity in samples of Ct > 25, where an evaluation of 122 CE-marked SARS-CoV-2 Ag RDTs found only 20.8% (20 tests) showed a detection rate >75% in samples of Ct range 25–30 [47]. Commercial LFTs are proprietary and have many differing parameters that can impact test performance, one of these being the antibody pairs. Therefore, we also evaluated the expected performance of our in-house AuNP assay with the same antibody pairs to serve as a direct comparator against FND performance.

For comparison, the clinical sensitivity of the AuNP assay was calculated using the fitted exponential model of the recombinant antigen data (Fig. 2b) to extrapolate the AuNP limit of detection in terms of FND signal (Eq. (2)). This yielded a fitted AuNP clinical

sensitivity of 56.6%, where only 2/10 moderate viral load ( ~ 10⁴-10⁵ copies/mL) samples may be detected (Fig. 3b, Supplementary Table 5). The fitted AuNP assay threshold of ~10⁶ copies/mL aligns well with sensitivity of commercially available LFTs reported in literature[30–32] and independent evaluations such as FINDDx (https://www.finddx.org/covid-19/). These relative thresholds, and clinical sample viral load concentrations were taken together with infection dynamics reported in literature[25,48,49] to calculate the FND diagnostic advantage in terms of days before diagnosis. Using human challenge data[25], the patients could be diagnosed 2.0 days earlier, on average [95% credible interval: 1.8 to 2.1, P(days$_{FND}$ > days$_{AuNP}$) < 10⁻¹⁰ (25 standard errors from zero)] using the FND assay, compared to AuNPs (Fig. 4a, Supplementary Fig. 9). The FND assay could detect 95% of patients with 95% confidence no later than 3.5 days, compared to 6.2 days for the AuNP assay, a diagnostic advantage of 2.8 days (Fig. 4a, b). The human challenge SARS-CoV-2 infection data- where a person is inoculated with the virus- is the only study format where the true day of initial

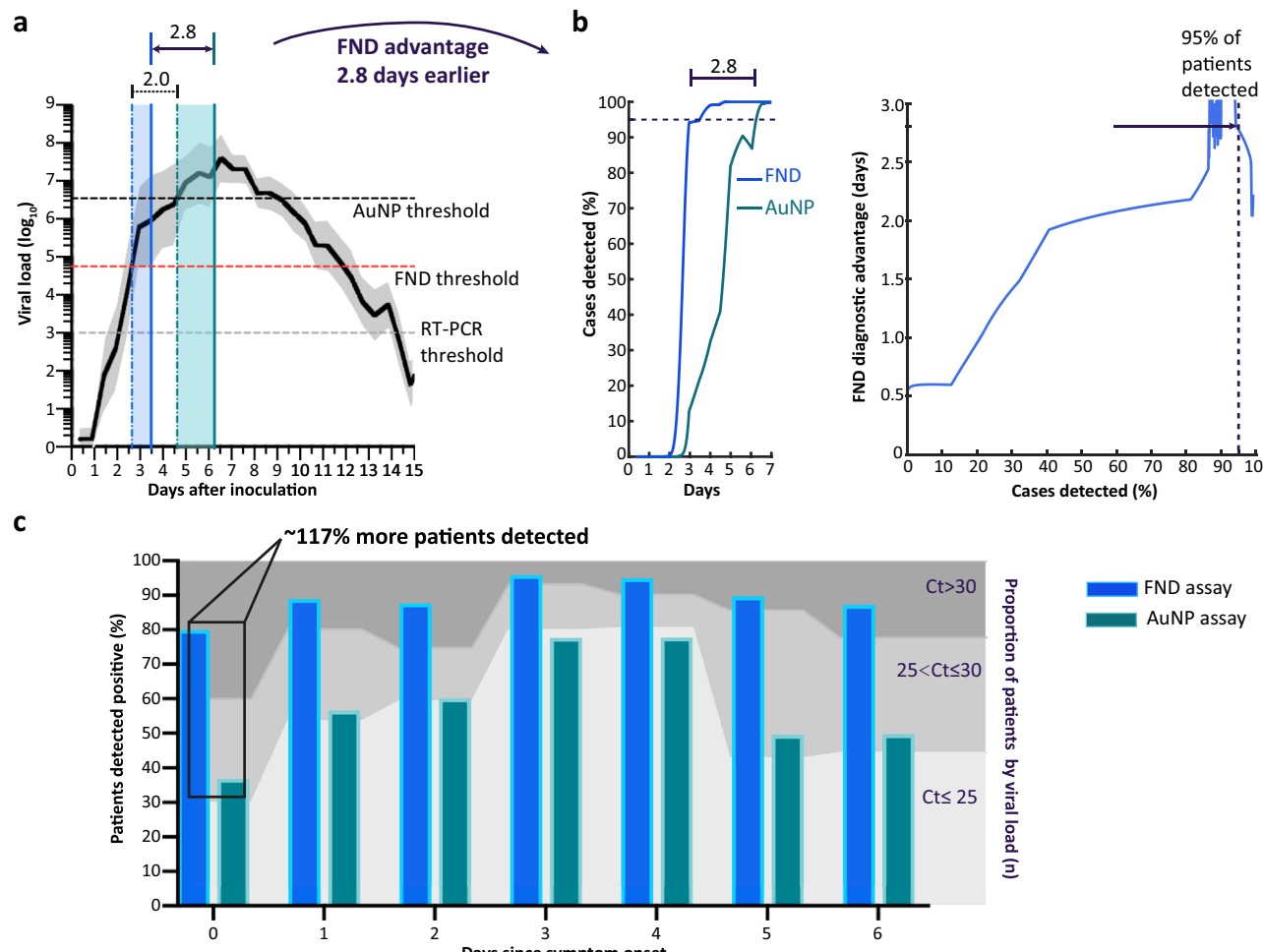

**Fig. 4 | Modelling diagnostic advantage of FND assay compared to AuNPs.**
**a** Viral load infection dynamics adapted from Killingley et al.[25] human challenge trial tracking viral load over the first 15 days of infection. Black line indicates mean patient viral load over time with grey region showing the 95% credible interval, using combined uncertainty from the assay thresholds and distribution of patient viral loads. The horizontal dashed lines represent mean assay thresholds at $10^3$ copies/mL for RT-PCR (grey), $5.6 \times 10^4$ copies/mL for FND (red) and $3.5 \times 10^6$ copies/mL for AuNP (black). The mean day of initial detection was 2.6 days [95% credible interval: 2.6–2.7] with the FND assay (vertical blue dashed line), and ~4.6 days [95% credible interval: 4.4–4.8] with AuNPs (vertical green dashed line). Taking the difference in means days shows that the FND assay can 2 days earlier [95% credible interval: 1.8-2.1], on average (unpaired two-tailed t-test, p-value < 0.0001, T = 25). The vertical shaded regions represent the gap between the detection of the 50th percentile viral load (mean) patients (vertical dashed lines) and 95th percentile patients with 95% confidence (vertical solid lines). We also show the FND assay could detect patients within 0.6 days of RT-PCR[25] (**b**) The left plot shows the

percentage of cases detected with FND and AuNP assays over time, shown as solid lines. The dashed line indicates 95% of cases detected. The FND assay will detect infection in 95% of patients no later than 3.5 days compared to the AuNP assay ≤6.3 days, a 2.8 day diagnostic advantage for detection with FNDs. The right plot shows the difference between the two assays in days (diagnostic advantage, solid line) by percentage of cases detected. The FND assay is more sensitive, so is increasingly advantageous as you get to higher percentages of cases detected (including more low-viral load patients). The dashed line indicates 95% of patients detected. **c** Line graph shows data adapted from Frediani et al.[48], representing the proportion of patient samples within ranges Ct ≤ 25, 25 < Ct ≤ 30, and Ct > 30 on each day from symptom onset (day 0). The bar graph shows the sensitivity of each assay weighted by proportion of high, moderate, and low viral load samples detected each day. Compared to the AuNP assay (green), the FND assay (blue) could detect ~117% (2.2-fold) more patients on the first day of infection (symptom onset) and ~57% (1.6-fold) more patients the second day. Source data are provided as Source Data file.

infection is known, independent of symptom onset, but is limited by the sample size of $n = 18$. Our sensitivity analysis was then extended to a larger clinical study (Frediani et al.[48]) that provides patient data ($n = 338$) in grouped Ct ranges per day following symptom onset. This suggests that ~117% (2.2-fold) more patients could be diagnosed with the FND assay on the first day of symptom onset than the conventional AuNP assay (Fig. 4c), an increase of 43 percentage points in the total patients. Real-world testing involves individuals at various stages of infection, so we also estimate the impact at the population level on a single testing day using the respective assay sensitivity across all viral loads, independent of individual infection stages. This analysis suggests that, at the peak of the Omicron wave in the U.K. (Jan 2022) the sensitivity of the FND assay is estimated to detect up to 69,400 more

patients in single day than our conventional AuNP assay (Supplementary Fig. 10). This approach provides an estimate of the number of additional patients that could be detected in a population-wide setting given improved sensitivity of the FND assay.

This work demonstrates a high level of concordance with RT-qPCR. However, interpretation of antigen sensitivity in low viral load samples using RT-qPCR Ct cut-offs should be taken with caution as (i) Ct values vary depending on the assay[50,51], and (ii) RT-qPCR can suffer from extended RNA positivity where thresholds for infectiousness in relation to Ct values has yet to be fully established[52]. Notably, Killingley et al.[25], reported quantifiable virus by RT-qPCR was still present by day 14 necessitating extended quarantine. In contrast, viable virus by cell culture, often used as a better indicator of infectiousness, showed viral

clearance by day 10.2 on average and no later than day 12. This supports compounding evidence that Ag-LFTs are good determinants of infectiousness, and more closely aligns with the FND assay thresholds where our fitting shows latest detection at day 12 (Fig. 4a). Therefore, 100% sensitivity across the whole range of RT-qPCR Ct values may not be desired. The reduction in sensitivity in low viral load samples (<10⁴ copies/mL) may be beneficial for differentiating active viral infection and residual circulating antigen. Further standardisation is needed on SARS-CoV-2 antigen test sensitivity and the minimum viral load needed to determine infectiousness.

Interpretation of sensitivity in SARS-CoV-2 clinical studies depends on sample selection and assay thresholds (e.g. evaluations including sample pools skewed to high viral loads can cause overestimation of test sensitivity than would be achieved in the general population). The initial random selection of samples in this study, from hospitalised patients to travellers, is somewhat representative of samples found in the general population. The second set of samples targeting Ct > 25 may skew the sample pool to the low viral end, suggesting an underestimate of overall clinical sensitivity. In addition, although this work did not find the swab type (nasopharyngeal and nasal only) had a significant impact on device performance (Supplementary Fig. 8), variability in viral load and resulting Ag-LFT performance have been reported in literature[25,30,53]. Further standardisation is needed to determine prevalence within these grouped Ct sample pools and swab type should be consistent so that pre-clinical device performance evaluations accurately depict performance in the field. Accurate determination of thresholds set by the negative samples can also impact test sensitivity and specificity. Our negative sample size of $n = 50$ gives a standard error of the mean of ±14% of σ, where σ is the true standard deviation of negative signal values, and standard error of σ of ±10% of σ. This is sufficiently small to accurately estimate the threshold. The FNDs high sensitivity also presents a promising avenue for early diagnosis in other diseases, like influenza where early administration of Tamiflu within 48 h of symptom onset can significantly improve health outcomes[54], and future paradigms such as HIV viral load self-monitoring on a LFTs[55].

Future work involves moving towards a cassette-based lateral flow test platform with integrated reagents. Initial experiments (detailed in Supplementary Information) demonstrate adequate reaction kinetics required for an integrated device where capture and detection of reagents typically occurs during rehydration of the conjugate and flow up the membrane (Supplementary Fig. 11). The lateral flow strip readout is a key consideration for fluorescence-based assays. More specific to our FND-based modulated platform, we considered the dielectric effect from a wet lateral flow strip on the resonator and subsequent lock-in result to allow for immediate read-out. The effect of wetting of the lateral flow showed a ~40% reduction (small compared to the LOD uncertainty and comparable to the strip-to-strip coefficient of variation of 18.3%) in the lock-in signal with immediate read-out after LFT running compared to after 20 min (Supplementary Fig. 12). However, this effect can be accounted for by normalising to the test membrane, or control line, or using a tuneable resonator to mitigate shifts in resonant frequency as the strip dries. The small size of the resonator and high brightness of FNDs allow for this system to be integrated into a portable, cost-effective smartphone fluorescent reader with estimated production cost of our prototype device at ~£912 (Supplementary Table 6).

To conclude, we demonstrate that spin-enhanced FNDs are a highly sensitive, specific and versatile platform for antigen detection, achieving a ~1000-fold improvement over gold nanoparticles using identical capture antibodies, giving a 2.8-day diagnostic advantage to detect virus in 95% of patients, and could have detected up to 69,400 more cases at the peak of the Omicron wave. Spin-enhanced FNDs showed 95.1% sensitivity in clinical samples (Ct ≤ 30) and 100% specificity compared to RT-qPCR. The sensitivity achieved is comparable to

high end Quanterix biomarker discovery platforms at a fraction of the cost. Our findings demonstrated the utility of nanodiamonds in a large blinded clinical study, laying the foundations for future research to explore antigen detection in clinical samples with high background and assays with low nonspecific binding. Future research is needed to miniaturisation of optical and modulation components into a portable fluorescent reader to improve access for low-resource and point of care settings, for example A&E, pharmacies, and home monitoring (e.g. disease or drug). Additionally, a larger clinical study is needed, and extension to other diseases, both communicable and non-communicable.

## Methods
### Ethics statement
We confirm all data collected in this study complies with ethical requirements. The UCLH Governance Committee has approved the study performed under the UCLH HTA license for use of residual samples for development of diagnostic assays (IRB no. NDU-VIR_131/13122022). All samples where pseudo-anonymised at source and tested prior to discard. No personal identifiable data including sex, gender, race, or ethnicity information were collected by the research team and were not disclosed to or considered by investigators in this study, therefore no individual informed consent was required for this study (UCL Infection DNA Biobank 2022 (IRAS ID 320050)).

### Statistics & Reproducibility
Biological replicates are defined as clinical samples from different patients of the same condition (e.g. negative controls). A test replicate is defined as a distinct test strip. Multiple tests could be run from a single biological (synthetic or clinical) sample. A sample size of $n = 3$ distinct test replicates at 7 (or more) sample concentrations was selected based on previous studies using similar methodologies. This provides sufficient data for accurate curve fitting and statistical analysis. Key findings on characterisation and analytical limit of detection were replicated in independent experiments following reported methods. Replication studies were successful. The inclusion of $n = 103$ clinical samples provides sufficient statistical power and considers variability within the population. Clinical evaluation with residual samples could not be replicated due to insufficient sample volume. Residual clinical samples were randomly selected by individuals at UCLH to include a random selection of positive and negative samples, broadly representative of the normal population. Blinded residual clinical samples were assigned a unique identification number by individuals at UCLH for transfer to investigators for initial data collection and analysis. Post-analysis, investigators received unblinded sample identification number with corresponding clinical results. No data was excluded in the analyses of this study.

### Materials and reagents
FortéBio Octet Red96 Streptavidin sensor plates (Part No 18-5019) and Kinetics buffer 10X (Part No 18-1092) obtained from Sartorius. 600 nm polyglycerol-coated fluorescent nanodiamonds (FNDs) were purchased from Adámas Nanotechnologies (custom purchase, 1 mg/mL in DI H₂O, ~3 ppm NV). Triton X-100 lysis buffer, pH 7.4 (Cat # J62289, Alfa Aesar) purchased from Thermo Scientific. Polystreptavidin test line nitrocellulose half-strips were purchased from Global Access Diagnostics (UK). N,N-Dimethylformamide (Cat#227056), N,N-Disuccinimidyl carbonate (Cat#43720), IGEPAL CA-630 (Cat#I8896), Trizma hydrochloride (Cat#T3253), Trizma base (Cat#T6066), Casein Hammarstein Bovine (Cat#E0789) were purchased from Sigma Aldrich. NaCl (Cat#27810.295) and Tween20 (Cat#437082Q) purchased from VWR chemicals. SARS-CoV-2 nucleocapsid antibodies (Cat#: 40143-R001, clone ID 001, RRID AB_2827974; 40143-R001-B, clone ID 001,Lot#HP15JA2004; 40143-R004, clone ID 004, RRID AB_2827975; 40143-R040, clone ID 040, RRID AB_2827976; 40143-

MM08, clone ID 08, Lot# MA14DE0202), SARS-CoV-2 biotinylated recombinant nucleocapsid protein (Cat# 40588-V08B-B), and MERS-CoV (Cat# 40068-V08B), HCoV-OC43 (Cat# 40643-V07E), HCoV-299E (Cat# 40640-V07E), HCoV-NL63 (Cat# 40641-V07E), HCoV-HKU1 (Cat# 40642-V07E) recombinant nucleocapsid protein were purchased from Sino Biological. SARS-CoV-2 recombinant nucleocapsid protein was provided by Peter Cherepanova (Francis Crick Institute) and Prof Ciara O'Sullivan at Universitat Rivira i Virgili (URV). AbCR3009 and AbCR3018 were provided by Laura McCoy at UCL Division of Infection and Immunity. The following reagents were obtained through BEI Resources; NIAID, NIH: SARS-Related Coronavirus 2, Isolate USA-WA1/2020, Gamma-Irradiated, NR-52287, contributed by the Centers for Disease Control and Prevention, and NIH: SARS-Related Coronavirus 2, Isolate hCoV-19/USA/GA-EHC-2811C/2021 (Lineage B.1.1.529; Omicron Variant), Gamma-Irradiated, NR-56496, contributed by Mehul Suthar. 40 nm citrate gold nanoparticles (AuNPs) were purchased from Nanocomposix (SKU: AUCR40-5M). ACON Biotech FlowFlex SARS-CoV-2 Rapid Antigen Test (Self-Testing) 25 Tests (Ref L031-118Q5 PZN-17522027). Clinical Nasopharyngeal swab samples were obtained from UCLH. N1 2019-nCoV RUO kit (Integrated DNA Technologies Cat#10006713). Thermofisher TaqPath™ 1-Step RT-qPCR Master Mix, CG (Cat# A15299). SARS-CoV-2 synthetic RNA positive control (Twist Bioscience, control 51 cat#105346). In-house lateral flow test assembly: Nitrocellulose membrane CN95 (UniStart Sartorius part no. 1UN95ER050025WS), backing card 60mmx300mm purchased from Kenosha (KN-PS1060.45), sink pad: cellulose fiber 20 cm x 30 cm CFSP223000 (Millipore)

### Biolayer Interferometry for antibody binding kinetics
Biolayer interferometry (Fortebio Octet Red, data acquisition 8.2) was used to analyse the antibody binding kinetic parameters. The assay was performed using a streptavidin-coated biosensor (Octet® Streptavidin (SA) biosensor, P/N 18-5019). A volume of 200 µL sample or kinetics buffer (P/N 18-5032) was added to a 96-well plate (Greiner bio-one) for each baseline, loading, association, and dissociation step. Biotinylated nucleocapsid protein (Sino biological Cat# 40588-V08B-B) was loaded on the SA sensor at 1 µg/mL. Following a baseline step, the biosensors moved to wells containing a 2-fold dilution series of the target nucleocapsid antibody (CR3009, CR3018, 40143-R001, 40143-R004, 40143-R040, 40143-MM08) diluted in kinetic buffer for the association step. Antibody concentrations ranged from 20 µg/mL to 0.001 µg/mL with each sample performed in triplicate. The binding curves were analysed in MATLAB using a 1:1 binding kinetics model (MATLAB code provided in data repository[56]).

### Fluorescent nanodiamond functionalisation
Polyglycerol (GF)-coated FNDs 600 nm were conjugated to antibodies using disuccinimidyl carbonate (DSC). 100 µL of 600 nm FND-PG at 1 mg/mL (Adámas Nanotechnologies) were added to a protein low-bind tube and placed in the bath sonicator at high power for 5 min. The particles are then centrifuged at 376 x g for 4 min, immediately followed by a 30 s spin at 21,130 x g to concentrate the particles to a pellet. The supernatant is removed and resuspended in anhydrous N,N-dimethylformamide (DMF, Sigma-Aldrich, Cat#227056). The suspension is mixed well and sonicated for 2 min at 100% power. The centrifugation and wash steps are repeated 3X to remove water. After the last centrifugation step, the particles are resuspended in 100 µL of 50 mg/mL DSC in DMF and placed on the thermoshaker for 3.5 hrs at 300RPM and 25 °C. Next, excess reagents are removed by a centrifugation and 3X wash steps to resuspend DMF. After the last centrifugation, the activated particles are resuspended with 100 µL of AbMM08 (2.71 µg) diluted in DI water. The reaction tube is placed on the thermoshaker for 15–17 hrs at 250RPM and 25 °C. Subsequently, the reaction is quenched with with the addition of 10%(wt/v) of 1 M

Tris-HCl pH 7.5 and placed on the thermoshaker for 30 min. The particles are centrifuged at 21,130 x g for 1 min and washed with DI water to remove unbound reagents, this step is repeated three times. The final resuspension is in 100 µL 0.1% (wt/v) BSA in PBS. The functionalised FND concentration was then measured by fluorescent intensity against a standard curve of the stock 600nm-FND-PG solution. The dilution series was prepared in a 96-well plate and fluorescent intensity was measured on the spectrophotometer (CLARIOstar, BMG, MARS version 4.20) with 550 nm excitation and 675 nm emission. A linear regression was fit to the fluorescence intensity of the stock dilution series against particle concentration (M) and interpolated to find the molar concentration ($c_p$) of functionalised FND particles using Eq. (1), where $d$ is particle diameter, $\rho$ is density of diamond, and $N_A$ is Avogadro constant.

$$C_P(M) = \frac{c_p(\mathrm{mgmL^{-1}})}{d^3(\mathrm{nm}^3)} \times \frac{10^3}{\frac{\pi}{6} \times \rho(\mathrm{mgnm^{-3}}) \times N_A(\mathrm{mol^{-1}})} \tag{1}$$

### Gold nanoparticle (AuNP) functionalisation
The parameters for antibody physisorption were determined using 40 nm citrate AuNPs (Nanocomposix) at OD1 and swept across antibody concentrations from 8.7 to 70 µg/mL, in buffers ranging from pH 7.7 to 9.0. A salt stress test was applied and particle absorption spectra was measured using plate reader (CLARIOstar, BMG, MARS version 4.20) to check for aggregation and absorption properties. A 1 mL aliquot of AuNPs was mixed with 200µL of AbMM08 at 35 µg/mL in borate buffer pH 8. This mixture was incubated at room temperature for 1 hr at 650RPM (Thermoshaker). Blocking was performed by adding 100 µL of 1 mg/mL BSA in $H_2O$ and left to shake for 30 min. The mixture was washed by centrifuge (3 washes at 10,000 x g for 10 min) and resuspended in 5% BSA + 0.05% Tween20 in PBS.

### Particle characterisation
Dynamic light scattering (Anton Paar, Litesizer 500, Kalliope version 2.22.1) was used to characterise the hydrodynamic diameter and monodispersity- polydispersity index (PDI) of the nanoparticles. Samples for scanning electron microscopy (SEM) were prepared by drop casting the 600 FND-PG in 12.5 mM $MgCl_2$ in PBS and dried for 30 min. The addition of 12.5 mM $MgCl_2$ improves particle adhesion to the substrate surface. Samples were imaged on the focused ion beam Zeiss XB1540 microscopy system. Quantitative analysis of particle size was performed across 5 SEM images with a total count of 156 FNDs (ImageJ 1.53 u). The width and diagonal (longest dimension) of each particle were measured with reference to the SEM scale bar. These measurements were then used to calculate particle size, assuming an approximate spherical shape. The 600 nm FND excitation spectrum was measured on a plate reader (CLARIOstar, BMG, MARS version 4.20) by sweeping excitation wavelengths from 320 nm to 640 nm with the emission filter set to 675 nm. The emission spectrum was measured by a widefield fluorescent microscope coupled to an Optosky ATP2000P-340-850-100 spectrometer at 2 ms integration time.

### Running buffer formulation
A clean beaker was placed on a magnetic stir plate with a magnetic bar for continuous stirring. Formulation was prepared at 5X concentration of IGEPAL CA-630, Casein, and Tween20 in 50 mM Tris-HCl + 100 mM NaCl base buffer. 50 mM TRIS-HCl solution was made up to pH 8 mixing appropriate amounts of Trizma Hydrochloride (MW: 157.60 g/mol) + Trizma Base (MW: 121.14 g/mol) (mg) in DI water, continuously stirring until fully dissolved. NaCl was added at 100 mM in final solution volume, continuously stirring until fully dissolved. The 50 mM Tris-HCL + 100 mM NaCl buffer was split into two batches, one

stored at 2–8 °C until further use, and the second batch was used to prepare the 5X buffer. While continuously stirring, 10% w/v IGEPAL CA-630 was slowing added to the mixture, followed by 0.25% w/v Tween20. Next, 4% w/v Casein (Hammarstein bovine, Sigma, Cat#E0789) was added, and the temperature was raised to 40 °C monitored by a temperature probe, covered, and continuously stirred for 24 hrs. Following, the buffer was aliquoted and frozen at -20 °C. The 5X buffer was thawed and diluted 1:5 in base buffer (50 mM Tris-HCL, pH8 + 100 mM NaCl) to make a 1X solution and stored at 4 °C for lateral flow testing.

### Lateral flow test line printing
CN95 nitrocellulose membrane at 2.5 cm width (Sartorius) was laminated onto backing card (Kenosha) with 2.2 cm absorbent pad (Millipore) and cut into 15 mm card segments for printing (Biodot AD1520 dispenser, software Link2020, AxSys). Antibodies were diluted from stock concentration to 1 mg/mL in 1X PBS for printing at 1μL/cm, with test line printed 1 cm from the bottom of the nitrocellulose membrane. Following printing, cards were placed in the oven to dry for 2hrs at 37 °C. Strips were cut with strip cutter (ZQ2002, Shanghai Kinbio Tech. Co., Ltd.) at 3 mm width. Strips were stored in a desiccated bag until use.

### FND lateral flow testing
The lateral flow assays were ran using commercially available strips with polystreptavidin printed test lines (Global Access Diagnostics). FND-PG-AbMM08 was diluted in PBS to concentration of 26.4 fM. For the direct bind format, 5 μL of the FNDs + 50 μL of biotinylated nucleocapsid protein diluted in running buffer (150 mM Tri-HCL pH 8, 2% IGEPAL CA-630, 100 mM NaCl, 0.05%Tween20, 0.8%Casein) was added to wells in a 96-well plate and allowed to bind for 10 min. The strips were added to each well and allowed to run for ~15 min. Strips were allowed to dry before read-out on the microscope described in the following section. For the sandwich format assay, FND-PG-AbMM08 was diluted in PBS to concentration of 26.4 fM and secondary biotinylated AbR001 was diluted in PBS to 513 nM. 5 μL of the FNDs + 9μL of sample (nucleocapsid protein or gamma-irradiated virus) diluted in running buffer + 1 μL of bAb01 (513 nM) was added to wells in a 96-well plate and allowed to bind for 10 min. Strip were added to the wells and run for 15 min, followed by test line read-out of dry strips.

### FND test line analysis
The lateral flow strips were imaged using a fluorescence microscope (Olympus BX51) with a 550 nm green LED excitation light source (CoolLED pE-4000), a filter cube with an excitation filter (Semrock-500nm bandpass, 49 nm bandwidth), a dichroic mirror (Semrock-596 nm edge), and 593 nm long-pass emission filter (Semrock). A 20x/0.4 BD objective lens was used. Images were recorded using a high-speed camera (Hamamatsu, ORCA-Flash4.0 V3) and HCImage Live software (Hamamatsu, v4.1.1.0) where the mean of each frame were calculated to give a time-series of mean pixel values. A voltage controlled oscillator (VCO) (Mini-Circuits-ZX95-3360 + ) and an amplifier (Mini-Circuits-ZX60-33LN + ) were connected to an antenna resonator and circuit board (Minitron, Rogers 4003c 0.8 mm substrate with 300 gm-2 copper weight) to generate the microwave field. The resonator was designed to match the frequency of ~2.87 GHz to produce desirable contrast of FNDs at the test line. The modulated signal is achieved by modulating the VCO input with a reference frequency generator at 4 Hz, using a 32.768 Hz crystal oscillator (Farnell, DS32KHZ) and 14-stage frequency divider (Farnell, CD4060BM). The fluorescence signal was modulated with a set modulation frequency ($Fm$) and the amplitude of the modulating signal was processed using a computational lock-in algorithm via MATLAB (see code in data repository[56]).

### Optically detected magnetic resonance (ODMR) measurements
An ODMR measurement was taken from a strong positive test line on the nitrocellulose strip to validate the fluorescence output from the FND NV- centres in the presence of a microwave field. A lateral flow strip with 600 nm FND-PG-AbMM08 bound to the test line in a direct bind format via biotinylated nucleocapsid protein (10 ng/mL) was placed under the microscope set-up with microwaves being supplied using a (SynthUSB3) source through a stripline and a high-power amplifier (12 V, ZRL-3500 + , Mini Circuits) to sweep across frequencies from 1 to 3 GHz. A MATLAB script was used to plot the frequency intensity across different frequencies.

### Limit-of-detection analysis
A 2-fold serial dilution of recombinant nucleocapsid protein or SARS-CoV-2 gamma irradiated virus was ran on the half-strip dipstick sandwich format assay and the lock-in intensity values were used to determine the limit of detection. The LoD was plotted and calculated based on an exponential model fit using a MATLAB software tool developed by Miller et al.[42], which is available open source on Github (https://github.com/bensmiller/detection-limit-fitting/) and data repository Zenodo with identifier https://doi.org/10.5281/zenodo.13266258[57].

### Evaluation of clinical swab samples
Nasal and nasopharyngeal swab samples were collected from individuals at UCLH (including travelers, healthcare workers, or other patients) from Nov-Dec 2022 and June-Aug 2023. Ct values were initially determined following HSL in-house standard protocol. Residual samples were stored at -80 °C until further testing. Samples were randomly selected by UCLH for transfer and tested blinded on the FND Ag-LFT in-house. Subsequently, samples were unblinded for analysis and positive samples were re-evaluated in-house by RT-qPCR using N1 2019-nCoV RUO kit (Integrated DNA Technologies Cat#10006713) and 4X TaqPath 1-step RT-qPCR Master Mix (Thermofisher, Cat#A15299) on QuantStudio RT-qPCR system (ThermoFisher Scientific, Quant-Studio v1.5.1). 5-point 10-fold dilution series of SARS-CoV-2 synthetic RNA positive control (Twist Bioscience, control 51 cat#105346) was used for standard curve quantification to RNA copies/mL (Design&Analysis 2.6.0). For lateral flow testing, 40μL of swab sample in VTM was added to the well and mixed with 9μL of 5X running buffer + 5 μL of the FNDs (26 fM) + 1μL of AbR001-b (513 nM) and allowed to bind for 10 min. Strips were added to wells and ran for 15 min. The strips were allowed to dry before read-out described in the previous section.

### Estimating AuNP Ag-LFT clinical sensitivity using recombinant antigen fitted LoD
In order to calculate the clinical sensitivity of the AuNP assay, a relative threshold compared to FNDs was calculated. This threshold was taken as the FND signal for a sample with a recombinant antigen concentration corresponding to the fitted LoD of the AuNP assay. The signal y-value ($\log_{10}$) was calculated based on the following Eq. (2):

$$y = \log_{10}\left( a + b * \left( 1 - \exp\left( -\frac{10^{[LOD_{AuNP}] - 2}}{c} \right) \right) \right) \qquad (2)$$

where $y$ is the FND signal corresponding to the AuNP LoD, $LOD_{AuNP}$ is the AuNP LoD, and $a$, $b$, and $c$ are parameters fitted in Fig. 2a. The threshold was applied to clinical sample data (Fig. 3b) to estimate clinical sensitivity of the AuNP assay.

### Modelling clinical assay performance from infection dynamics of SARS-CoV-2 human challenge infection
Infection dynamics data from the SARS-CoV-2 human challenge trial[25] were used to calculate the expected clinical performance of the FND

and AuNP assay over the course of infection to estimate the time from infection to detection for each assay. The challenge infection data provides the true days of infection, independent of symptom onset. The data plot is extracted from Killingley et al.[25]. Figure 2a, which represents mean viral load and standard error of the mean in nasal swab samples taken twice daily from the participants ($n = 18$). The study found a median of -2.24 days [95% CI: 1.67-3.17] to initial quantifiable detected virus with RT-qPCR at a threshold set at $10^3$ copies/mL. We calculate the mean day of initial detection from patient samples RT-PCR at -2.08 days.

First, we used a symmetric Bayesian linear regression with errors-in-variables (x and y) to fit FND signal ($log_{10}$) and viral load ($log_{10}$(copies/mL))[58]. This used Gaussian prior distributions for the gradient and intercept, both with a mean of 0 and a standard deviation of 10. The log likelihood function uses the orthogonal residuals from the model, weighted by the variance in the orthogonal direction:

$$\text{log likelihood} = \sum_{i=1}^{n} \left[ -\frac{1}{2} ln(2\pi\sigma_i^2) - \frac{d_i^2}{2\sigma_i^2} \right] \quad (3)$$

where $d_i$, is the orthogonal residual, and $\sigma_i^2$ is the combined variance from uncertainty in x and y:

$$d_i = \frac{y_i - \hat{y}_i}{\sqrt{1+b^2}} \quad (4)$$

where $y_i$ is a measured y value, $\hat{y}_i$ is the y value for a given $x_i$ predicted by the model: $\hat{y}_i = a + bx_i$ with y-intercept $a$, and gradient $b$.

$$\sigma_i^2 = \frac{\sigma_{y_i}^2 + b^2 \sigma_{x_i}^2}{\sqrt{1+b^2}} \quad (5)$$

where $\sigma_{y_i}^2$ is the y-variance and $\sigma_{x_i}^2$ is the x-variance of a point ($x_i, y_i$).

This model was then used to calculate the equivalent copies/mL values and associated uncertainty for the FND signal LoD threshold, and relative FND signal for the AuNP LoD. The uncertainty was calculated by propagation of errors:

$$x_{est} = \frac{y_t - a}{b} \quad (6)$$

$$\sigma_{x_{est}} = \sqrt{\left(\frac{\sigma_{y_t}}{b}\right)^2 + \left(\frac{\sigma_a}{b}\right)^2 + \left(\frac{(y_t - a)\sigma_b}{b^2}\right)^2} \quad (7)$$

where $x_{est}$ is the estimated copies/mL for a FND signal threshold $y_t$.

The Metropolis-Hastings algorithm is used to sample from the posterior distribution[59].

The FND threshold was $5.6 \times 10^4$ copies/mL [95% credible interval: $4.2 \times 10^4$ to $7.5 \times 10^4$] and AuNP threshold was $3.5 \times 10^6$ copies/mL [95% credible interval: $2.3 \times 10^6$ to $5.1 \times 10^6$].

These thresholds were then plotted on the infection dynamics graph (Killingley et al.[25], Fig. 2a) and interpolated to find the estimated timepoint (days) to initial detectable virus across each assay (Fig. 4a). The day of initial detection for the patient with the mean viral load was calculated for both the FND and AuNP assay with 95% confidence intervals, accounting for error in the viral load thresholds. Following, the difference between the mean days with standard error was used to indicate the average window for earlier detection with FNDs. We then used the distribution of patient viral loads over time (Killingley et al.[25]) along with the uncertainty in viral load thresholds to calculate the percentage of detectable patients over time for each assay and, correspondingly, the FND diagnostic advantage in days against patient detection percentage.

## Calculating expected clinical assay performance by day of infection in a clinical symptomatic cohort

This analysis uses calculated sensitivity of FND and AuNP assays across a range of viral loads to show the approximate number of infected patients that could be detected per day. This data is adapted from Frediani et al.[48], that presented a large clinical study of viral loads in the days following symptom onset. The proportion of patients with high viral load samples (Ct ≤ 25), moderate viral load (25 < Ct ≤ 30), and low viral load (Ct > 30) sampled on each day was used to estimate the patients detected by the FND and AuNP assay per day. The calculated assay sensitivities by viral load, shown in Supplementary Table 5, were applied proportionally to the number of patients samples within the viral load range per day. This was used to determine the percentage of patients that would be detected with the FND and AuNP assay on each day of infection (Fig. 4c).

### Reporting summary

Further information on research design is available in the Nature Portfolio Reporting Summary linked to this article.

## Data availability

The Source data are provided with this paper. Larger datasets are available in Figshare with identifier: https://doi.org/10.6084/m9.figshare.29490068[56]. These are: the mean pixel data of each 750 image data set for each sample analysed for limit of detection analysis (Fig. 2b,d; Supplementary Fig. 6) and all clinical samples tested (Fig. 3), and raw data for antibody BLI kinetics plot (Fig. 2a, Supplementary Fig. 2, Supplementary Table 1). We have deposited the mean pixel data (simply the mean of each image, which is used for all analysis) rather than raw images (raw images are never used directly) for all samples tested in this work due to the large file size (750 images per replicate). The raw image data can be directly provided upon request of the corresponding author(s). We will process all requests within 2 weeks. Source data are provided with this paper.

## Code availability

The MATLAB software for fitting robust detection limits (LODs) and confidence intervals to serial dilution data is available on Github with Zenodo indentifier: https://doi.org/10.5281/zenodo.8346293[57]. The code used for FND lock-in analysis, antibody binding kinetics and Bayesian analysis are available in Figshare with identifier: https://doi.org/10.6084/m9.figshare.29490068[56].

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

## Acknowledgements

This work was funded by the i-sense EPSRC IRC in Agile Early Warning Sensing Systems in Infectious Diseases and Antimicrobial Resistance (EP/R00529X/1) to R.A.M., A.T.D., D.H., F.D., M.M.; The i-sense Next steps award (EP/R018707/1) to D.H.; A London Centre for Nanotechnology Departmental Studentship to A.T.D.; Wellcome Trust 224071/Z/21/Z to B.S.M.; The EPSRC Digital Health Hub for AMR (EP/X031276/1) to D.H., E.N. We thank The National Institute for Health Research University College London Hospitals Biomedical Research Centre and the UCLH NHS Foundation Trust (J.B. and E.N.). We thank Peter Cherepanov (Francis Crick Institute) for providing recombinant nucleocapsid protein and Laura McCoy (UCL) for providing antibodies CR3009 and CR3018.

## Author contributions

B.S.M. is the primary contact for this paper. A.T.D., B.S.M and R.A.M. conceived the research and led the study; A.T.D. designed and executed all experiments; B.S.M. co-supervised A.T.D., and the analysis of Forte bio kinetic data, LFT LOD analysis and statistics using bespoke software, and modelling clinical threshold Bayesian analysis; D.H. co-supervised A.T.D. and helped with the RT-qPCR experiment and analysis; F.D. and M.M. assisted with nanodiamond optical characterisation; E.N. and J.B. provided clinical expertise and clinical samples; C.K.O provided recombinant protein; L.E.M. provided antibodies; A.T.D., B.S.M., F.D. and R.A.M. drafted the manuscript; and all authors reviewed and revised the manuscript.

## Competing interests

The remaining authors declare no competing interests.
Benjamin Miller and Rachel McKendry have submitted patent applications to the UK Intellectual Property Office (application number GB 1814532.6; priority application filed 6 September 2018), to the World Intellectual Property Organization under the Patent Cooperation Treaty (application number PCT/GB2019/052474), to the European Patent Office (application number EP 19766322.2 A), and to the United States Patent and Trademark Office (application number US 17/273,641) per-taining to the fluorescence-based in vitro diagnostic assay methods of this work. The remaining authors declare no competing interests.
