## [Transparent Peer Review file · Nature Communications]

Quantum-enhanced nanodiamond rapid test advances early SARS-CoV-2 antigen detection in clinical diagnostics

Corresponding Author: Professor Rachel McKendry

Version 1:

Reviewer comments:

Reviewer #1

(Remarks to the Author)

This is a very interesting, detailed and well-written manuscript. The quantity of work behind this manuscript is remarkable, especially the data analysis is very complete and trustable. We propose this work to be accepted after some minor revisions described below:

1. In the text it is often mentioned that samples were collected using nasal swabs (e.g. line 22), however in the supporting information it is stated that samples were nasopharyngeal (end of section 1a). Nasal and nasopharyngeal samples are technically not the same. Please check which term is more correct and revise the text.
2. In line 70: TB abbreviation is mentioned for the first time (we understand it is for tuberculosis). As the abbreviation is not used again in the text, we suggest writing "tuberculosis" instead.
3. Could be possible to show in a histogram the size dispersion of the 600nm FNDs? (in relation to extended data Fig.1a).
4. In line 138: How were FNDs immobilized on test line for this case? Is it as shown on extended data Fig.1b or as in Fig.1d?
5. In supplementary information table 1, we would recommend to include information about the supplier of the antibodies or indicate if they were made in-house. It could probably be included in the first column, together with the antibody code.
6. In line 199: Authors mention that to reduce antibody concentration on TL they use strips with pre-printed streptavidin. In which percentage can they reduce the antibody concentration in comparison to non-streptavidin-printed strips? Could the authors add a reference supporting this?
7. In extended data figure 2a not all error bars are well-visible.
8. Why authors did not consider adding a control line on FND-LFTs? Please explain.
9. What about using a conjugate pad for the FNDs? May you explain?
10. What is the purpose of adding 12.5mM MgCl₂ to PBS buffer during SEM characterization?
11. In the methods section, on FND lateral flow testing, would not have been better to immobilize the secondary biotinylated antibody on the streptavidin test line, instead of mixing it with the running buffer? Are the KD of the antibodies not enough to perform the recognition on the test line?

Reviewer #2

(Remarks to the Author)

The manuscript "Quantum-enhanced nanodiamond rapid test advances early SARS-CoV-2 antigen detection in clinical diagnostics" presents the clinical application of a quantum-enhanced nanodiamond rapid test for early SARS-CoV-2 antigen detection. This study builds upon their previous work, "Miller, Benjamin S., et al. Spin-enhanced nanodiamond biosensing for ultrasensitive diagnostics. *Nature* 587.7835 (2020): 588-593.", which introduced the concept of nanodiamond-based lateral flow tests (LFTs) for rapid early-stage disease detection. While the clinical validation of this method is of interest, the manuscript primarily focuses on methodological validation rather than presenting a significant conceptual or technological advancement. As a result, its novelty is marginal in relation to the standards of Nature Communications. Additionally, several other concerns need to be addressed. Therefore, in its current form, the manuscript does not appear suitable for publication in Nature Communications.

1. The control line is missing in your nanodiamond-based LFTs. In LFT, the control line is essential for ensuring the validity and reliability of test results. Without a control line, it is impossible to determine whether the test functioned correctly or failed due to factors such as insufficient sample volume, reagent degradation, or improper sample application. Additionally, the

absence of a control line increases the risk of false positives, which may arise from non-specific binding, background noise, or improper flow, falsely indicating the presence of the target analyte.

2. Regarding limit of detection (LOD), existing studies on LFT-based SARS-CoV-2 rapid detection have already demonstrated highly sensitive performance. For instance, an LFT-based antigen detection method has reported a detection limit as low as 0.5 pg/mL (ACS Applied Materials & Interfaces, 2021, 13(34): 40342–40353). Given this benchmark, what specific advantages does your approach offer in terms of sensitivity, reliability, or clinical applicability? Additionally, your detection system involves a high-speed camera and devices for microwave source, which increase the cost of detection. The procedure also requires professional operators, which conflicts with the principles of LFTs, designed to be portable, rapid, and cost-effective.

3. Noise interference presents a significant challenge in LET-based detection, directly affecting the limit of detection (LOD). In Extended Data Fig. 3(c), in addition to the color change in the C and T lines, a faint red background is visible. This is likely due to the partial retention of gold nanoparticles on the NC membrane during the flow process. A similar issue may arise with FNDs, particularly given the relatively large 600 nm nanodiamonds used in this study. To mitigate this, further optimization of the test strip and improvements in the signal-to-noise ratio should be considered. Additionally, in Extended Data Fig. 2(b), noticeable inconsistencies in brightness and contrast are observed across the six test strip images. If these variations are due to differences in photography conditions, capturing all six samples in a single shot could help standardize image quality and improve comparability.

Reviewer #3

(Remarks to the Author)

In the manuscript titled “Quantum-enhanced nanodiamond rapid test advances early SARS-CoV-2 antigen detection in clinical 3 diagnostics”, DeCruz et al. demonstrated a spin-enhanced nanodiamond LFT with sub-pg/mL analytical sensitivity. Moreover, they demonstrated that this analytical sensitivity translated directly to high clinical sensitivity for SARS-CoV-2 with 100% specificity. This improved analytical sensitivity also improved time-to-detection relative to traditional gold nanoparticle based LFTs.

The highly-sensitive LFT described in this manuscript provides a significant advancement to the field of rapid diagnostic tests for Covid-19 and potentially other LFAs for settings in which readers are appropriate. The authors utilized appropriate and rigorous statistical methods, e.g. Dunnett’s test and ANOVA when examining cross-reactivity, and ANOVA when comparing antibody performance. The data is high-quality and presented clearly. Analytical tests were conducted in a relevant matrix (VTM). The samples for the clinical study were also chosen appropriately – the authors utilized symptomatic patients or those returning from travel instead of utilizing healthy controls for their negatives. Selecting additional samples near the LOD to more accurately assess the limit-of-detection was also an appropriate choice. The high specificity is due to choosing a threshold based on an ROC curve; however, this is not a criticism but rather an intrinsic benefit of utilizing this technology instead of a traditional visually read LFT.

The authors correctly identified limitations of their study; for instance, they advise caution in analyzing data that is based on CT values from two different studies (e.g. figure 4C and lines 330-332). They additionally note difficulties in establishing a representative clinical pool and correctly note their study population may lead to underestimating clinical sensitivity while other studies may tend to overestimate sensitivity. The inclusion of their own gold LFT performance helps to elucidate the effect of the FND approach within the same clinical pool. While the research is ultimately very promising, the utility will be limited until a point-of-care compatible reader is realized in a way that allows use in settings that benefit from LFA use vs other existing sensitive covid assays. In the interim, traditional LFTs will continue to be utilized despite their sensitivity limitations.

Major issues:

Methods: The decision to utilize LFAs in a format with all wet reagents needs to be included in the discussion. My understanding from the method is that wet biotinylated antibody, sample and FNDs are placed in a 96-well plate and allowed to bind for 10 minutes prior to running them on an LFT. In a traditional LFT, the sample would rehydrate the conjugate and bind during flow. Do the authors have references for this being an acceptable approximation? Second, please include some discussion as to the decision to allow the strips to dry prior to reading them. Typically, LFTs will be read within 10 to 15 minutes while the nitrocellulose is still wet – not dry. Finally, the impact of this device relies on making a low-cost portable reader. The authors state that is possible (line 237) but should add a line or two explaining the feasibility of this to the discussion.

Minor issues:

In lines 55-56, the authors not spin-enhanced FND LFTs achieve a fundamental LOD of 8.2×10^{-19} M, 94,000-fold more sensitive than conventional gold nanoparticles but in practice it is limited by nonspecific binding to NC. If known, it would be helpful for the authors to provide the sensitivity enhancement in practice.

In line 60-79, it is unclear as to why sensitivity has been a challenge with FND biosensors. The previous paragraph highlighted the high-sensitivity of this approach – why did previous research find relatively modest sensitivity (e.g. 1.94 ng/mL for SARS-CoV-2 by Hsiao et al.?)

600 nm particles are generally large for lateral flow applications. Does this lead to restrictions on the types of compatible nitrocellulose, i.e., faster-flowing, larger-pore variants?

In line 165, did the authors proceed with all four commercially antibodies? If so, please make this clear.

Although clinical performance shows that the assay performs exceptionally well, down-selecting antibody pairs based on performance against antigen from a single source can lead to bias. Cate et al (<https://pubs.acs.org/doi/10.1021/acsomega.1c01253>) showed marked difference in antibody performance when looking at nucleocapsid proteins from two different vendors and for those derived from a clinical pool.

Can the authors speculate on the reasons the FlowFlex assay showed 13x poorer sensitivity than the devices claim? We agree the comparison with an in-house gold assay is the more important result; however, 13x poorer sensitivity than the device literature is significant.

Given that infectious virus determined by culture corresponds to a level of roughly 10⁶ copies/mL, is there concern this highly sensitive test will continue to remain positive after the infectious period similar to RNA tests?

Both antibodies selected by the authors have been used in various publications regarding LFTs – the authors may choose to cite a couple of these to highlight that these are generally considered high-quality antibodies for this target

In the methods, line 391-393, please state the concentration of particles in addition to the volume which is already provided. This will make the results more reproducible for other investigators.

In lines 413-418 the sandwich format is described. The authors state the sample, FNDs and bAb01 are added and allowed to bind for 10 minutes. Is the LFT then placed in and allowed to run for 15 minutes similar to the direct bind format? As written, it is unclear what happens after the 10 minute incubation.

Please include information as to how the samples were run in the “evaluation of clinical nasal swab samples”. Specifically, were both antibodies added to the well with a sample, allowed to bind for 10 minutes, and then run up the LFT? Was the LFA allowed to dry prior to imaging?

In supplementary information table 1 please indicate what the range in brackets refers to — if it is 95% Cis please note this

Overall, the paper presents an exciting clinical demonstration highlighting the utility of FND based LFTs. We recommend accepting the paper pending minor edits.

Reviewer #4

(Remarks to the Author)

Reviewer #5

(Remarks to the Author)

Version 2:

Reviewer comments:

Reviewer #1

(Remarks to the Author)

The authors have replied to all or comments and kindly considered all our suggestions. We are very satisfied with the current version of the manuscript and we have no additional comments. Therefore, we can recommend its acceptance in the journal.

Reviewer #2

(Remarks to the Author)

The authors identify the core innovation of this work as the translation of a previously proposed spin-enhanced LFT model into clinical applications. They conducted a series of experiments to evaluate its practical usability and reported encouraging results, which contribute to advancing the commercialization or industrial application of LFT technology. However, whether

the application of an existing technical model to clinical settings meets the innovation threshold expected by Nature Communications is not straightforward to assess from my disciplinary perspective. The editorial team, being more familiar with the journal's criteria for novelty, is in a better position to make this judgment.

With regard to the authors' response, the following points may benefit from further clarification or revision:

1. If the authors consider the clinical translation of the technique to be the primary innovation, the current abstract may not fully reflect this emphasis. While it provides a detailed account of clinical results and their potential implications, it gives limited explanation of how the spin-enhanced LFT ensures specificity and sensitivity in a clinical context. Briefly outlining the key methodological principles would improve clarity.

2. The table listing general cost for detection methods appears not to be directly derived from the current study. Including such a table may not be appropriate unless the data specifically reflect the detection work described in this manuscript. A cost analysis focused on the method proposed in this work would be more relevant and informative.

Reviewer #3

(Remarks to the Author)

The authors answered all reviewer questions and comments satisfactorily. Great work!

Reviewer #4

(Remarks to the Author)

Reviewer #5

(Remarks to the Author)

Response to Reviewers

Manuscript ID: NCOMMS-24-55657B

Title: Quantum-enhanced nanodiamond rapid test advances early SARS-CoV-2 antigen detection in clinical diagnostics

Authors: Alyssa Thomas DeCruz, Benjamin S. Miller, Da Huang, Max McRobbie, Felix Donaldson, Laura E McCoy, Ciara K. O'Sullivan, Johannes C. Botha, Eleni Nastouli, Rachel A. McKendry

We thank all the Reviewers for their feedback and constructive suggestions which has improved the manuscript. Reviewer 1 says our papers is 'very interesting, detailed and well-written manuscript. The quantity of work behind this manuscript is remarkable, especially the data analysis is very complete and trustable'. Reviewer 3 states that our 'highly-sensitive LFT described in this manuscript provides a significant advancement to the field of rapid diagnostic tests for Covid-19 and potentially other lateral flow assays (LFAs) for settings in which readers are appropriate' and note the 'rigorous statistical methods ... data is high-quality and presented clearly.' They state: 'Overall, the paper presents an exciting clinical demonstration highlighting the utility of FND based LFTs. We recommend accepting the paper pending minor edits.'

Below we systematically respond to each Reviewer's comments in turn.

Reviewer #1 (Remarks to the Author):

'This is a very interesting, detailed and well-written manuscript. The quantity of work behind this manuscript is remarkable, especially the data analysis is very complete and trustable. We propose this work to be accepted after some minor revisions described below'

We thank the Reviewer for their positive endorsement of our paper. We are delighted to respond to the minor comments.

Reviewer 1 Comment 1: 'In the text it is often mentioned that samples were collected using nasal swabs (e.g. line 22), however in the supporting information it is stated that samples were nasopharyngeal (end of section 1a). Nasal and nasopharyngeal samples are technically not the same. Please check which term is more correct and revise the text.'

Response to Reviewer 1 Comment 1: We thank the Reviewer and confirm that we have indeed used a combination of the sample types. This included 12/53 positive nasal swab samples, with 41/53 nasopharyngeal samples. We found no significant difference in fluorescent nanodiamond (FND) signals between the two sample types (t-test p-value=0.6457, T=0.4625, DF=51), and no significant impact on viral load between swab type (t-test p-value=0.9886, T=0.01432, DF=51) as shown in Response Figure 1. This analysis has been added as a box plot in a new figure in the supplementary information section 4b. We have also added this to the discussion with supporting references Killingley *et al.* (2022)¹, Glenn *et al.* (2022)², and Brümmer *et al.* (2021)³

Response Figure 1: (a) Box plot of FND results from RT-qPCR positive clinical samples stratified by nasopharyngeal swab (41/53) and nasal swab (12/53) type showing mean FND signal (black line). No significant difference between mean FND signal was observed (t-test p-value=0.6457, $T=0.4625$, $DF=51$). **(b)** Visual representation of FND results by clinical sample viral load and swab type showing distribution of viral loads in nasal swabs and nasopharyngeal swabs (no significant difference between mean viral load across sample type determined by t-test, p-value=0.9886, $T=0.01432$, $DF=51$).

Amendments to manuscript:

Amendments to the Main Text:

Abstract (Line 23): ‘Our blinded clinical study with 103 patient upper respiratory tract swab samples showed 95.1% sensitivity ($Ct \leq 30$) and 100% specificity benchmarked to RT-qPCR, with no cross-reactivity to influenza A, RSV, and Rhinovirus.’

Main text (Lines 272-276): ‘An important next step was to blindly evaluate the performance of the FND assay using clinical swab samples compared to gold standard RT-qPCR results (Ct value). 103 frozen samples, a combination of nasopharyngeal and nasal swabs, were received from UCLH collected from individuals (returning travellers and symptomatic patients) from Nov-Dec 2022 and Jun-Aug 2023 during the pandemic.’

Main text (Lines 289-291): ‘No significant differences in FND signals or viral loads were found between nasopharyngeal and nasal swabs (t-test p-values 0.6457 and 0.9886, respectively) as shown in Supplementary Information Fig. 3).’

Main text (Lines 364-369): ‘In addition, although this work did not find the swab type (nasopharyngeal and nasal only) had a significant impact on device performance (Supplementary information Fig. 3), variability in viral load and resulting Ag-LFT performance have been reported in literature^{25,53,54}. Further standardisation is needed to determine prevalence within these grouped Ct sample pools and swab type should be consistent so that pre-clinical device performance evaluations accurately depict performance in the field.’

Methods (Lines 499-500):

Evaluation of clinical swab samples

‘Nasal and nasopharyngeal swab samples were collected from individuals at UCLH (including travellers, healthcare workers, or other patients) from Nov-Dec 2022 and June-Aug 2023.’

Supplementary Information (Lines 107-115):

Response Figure 1 has been added to Supplementary Information Section 4b 'Clinical sample matrix stratification by nasal and nasopharyngeal swab type' as Supplementary Information Fig. 3.

Reviewer 1 Comment 2. 'In line 70: TB abbreviation is mentioned for the first time (we understand it is for tuberculosis). As the abbreviation is not used again in the text, we suggest writing "tuberculosis" instead.'

Response: We thank the Reviewer for identifying this error and have updated the manuscript accordingly.

Amendments to manuscript:

Main Text (Lines 71-73): 'In later work, Le et al.¹⁷, achieved a LoD of 0.02 ng/mL detecting ESTAT6 for tuberculosis diagnosis but direct antigen detection was limited by the use of cell-cultured samples.'

Reviewer 1 Comment 3. 'Could be possible to show in a histogram the size dispersion of the 600nm FNDs? (in relation to extended data Fig.1a).'

Response: We have analysed the size dispersion of the 600nm FNDs using two complementary methods. First, as shown in Extended Data Fig. 1c, dynamic light scattering (DLS) data provides the mean hydrodynamic diameter of the 600nm-FND-PG (628nm \pm 220). Following the Reviewer's suggestion we have now performed a quantitative analysis of the scanning electron microscopy (SEM) images, measuring individual FND sizes across 5 different SEM scans, with a total count of 156 FNDs. The width and diagonal (longest dimension) of each particle were measured with reference to the SEM scale bar. These measurements were then used to calculate particle size, assuming an approximate spherical shape. The resulting histogram and Gaussian fit (shown below in Response Figure 2) gives a mean of 607nm and confirms a narrow size distribution with 90% of particles (\pm 1.645 s.d. of the mean) falling between 505-708nm. This aligns well with the DLS measurements – we would expect the hydrodynamic diameter to be \sim 30nm larger than SEM due to the thickness of the polyglycerol layer⁴. This analysis further validates the monodispersity of the FNDs used in our lateral flow system.

Response Figure 2: Histogram with size dispersion of the 600nm FNDs based on counts from SEM scans, showing distribution across size ranges from 275nm to 925nm, with peak counts between 575-625nm particles of size 575-625nm. The distribution is fit to a gaussian curve (black line) with mean=606.9 nm (± 61.7).

Amendments to manuscript:

Extended Data Figure 1 (Lines 550-568): The histogram of size distribution and gaussian fit from SEM analysis has been added to Extended Data Fig. 1a alongside the SEM image.

Extended Data Fig. 1: FND characterisation and antibody pair selection

Extended Data Fig. 1: (a) Scanning electron microscopy (SEM) image (left) of polyglycerol-coated 600nm FNDs at magnification of 3.21 K X showing monodispersed particles. Call-out box shows particles at magnification 27.03 K X showing cubic morphology of particles. Histogram (right) with size dispersion of the 600nm FNDs based on counts from SEM scans, showing distribution across size ranges from 275nm to 925nm, with peak counts between 575-625nm particles of size 575-625nm. Distribution is fit to a gaussian curve (black line) with mean=607 nm (± 62) (b) Schematic of direct bind assay and plot of mean signal-to-noise ratio ($n=3$) with standard error of means using different antibodies functionalised to 600nm FNDs with FND-AbMM08 showing best SNR of 83, but with no statistically significant difference (p -value=0.71, ANOVA, $F=0.48$, $DF=11$, post-hoc Tukey's multiple comparisons). Data points represent the SNR of individual test replicates using the mean baseline ($n=3$). (c) Mean FND size distribution plot using dynamic light scattering showing particle intensity distribution (%) of both 600nm-FND-PG and antibody functionalised 600nm-FND-PG with hydrodynamic diameters of 628nm (± 220), and 643nm (± 140). (d) SNR of different antibody-printed test lines (Supplementary Information section 1c) with FND-AbMM08 conjugate. Positive sample was 10ng/mL recombinant protein ($n=3$). Error bars represent standard error of mean. Data points represent the SNR of individual test replicates using the mean baseline ($n=3$). Statistical significance determined by one-way ANOVA, $F=8.2$, $DF=8$, post-hoc Tukey's multiple comparisons, * denotes p -values <0.05 : 0.0251 and 0.0363 for AbR001 vs. AbR004 and AbR040, respectively. No significant difference observed between AbR004 and AbR040 (p -value=0.95).

Methods (Lines 439-443):

'Quantitative analysis of particle size was performed across five SEM images with a total count of 156 FNDs (ImageJ). The width and diagonal (longest dimension) of each particle were measured with reference to the SEM scale bar. These measurements were then used to calculate particle size, assuming an approximate spherical shape.'

Reviewer 1 Comment 4. 'In line 138: How were FNDs immobilized on test line for this case? Is it as shown on extended data Fig.1b or as in Fig.1d?'

Response: We confirm this was a direct bind format as in Extended Data Fig. 1b (see above response to comment 3 for reference and schematic shown in extended data fig. 1b). The immobilisation via biotinylated antigen format is described in the methods section for Optically Detected Magnetic Resonance (ODMR) measurements.

Amendments to manuscript:

Methods (Lines 485-489):

'A lateral flow strip with 600nm FND-PG-AbMM08 bound to the test line in a direct bind format via biotinylated nucleocapsid protein (10ng/mL) was placed under the microscope set-up with microwaves being supplied using a (SynthUSB3) source through a stripline and a high-power amplifier (12V, ZRL-3500+, Mini Circuits) to sweep across frequencies from 1 to 3 GHz.'

Reviewer 1 Comment 5. 'In supplementary information table 1, we would recommend to include information about the supplier of the antibodies or indicate if they were made in-house. It could probably be included in the first column, together with the antibody code.'

Response: We thank the Reviewer for this recommendation and have updated Table 1 with the antibody supplier information, including the original reference paper for antibodies CR3009 and CR3018, and added the label for reported 95% CI.

Amendments to manuscript:

Supplementary Information Table 1:

Capture Antibody (Manufacturer, ID)	K_D (nM) [95% CI]	k_{on} (sM) [95% CI]	k_{off} (s ⁻¹) [95% CI]
R001 (Sino Biological, RRID: AB_2827974)	0.34 [0.19-0.55]	3.0×10^5 [2.6-3.5]	2.0×10^{-6} [-33-37]
R004 (Sino Biological, RRID: AB_2827975)	0.22 [0.17-0.27]	3.7×10^5 [3.5-4.0]	8.5×10^{-5} [6.0-11]
R040 (Sino Biological, RRID: AB_2827976)	0.25 [0.12-0.49]	1.7×10^5 [-1.79-5.16]	2.3×10^{-4} [-12-16]
MM08 (Sino Biological, RRID: AB_2827978)	0.19 [0.091-0.36]	3.6×10^5 [3.2-3.9]	1.3×10^{-5} [-6.7-9.4]
CR3009 (In-house Dr Laura E McCoy, clone ID 03-009 ³)	3.3 [2.7-3.9]	9.7×10^4 [8.8-11]	2.6×10^{-4} [0.71-4.4]
CR3018 (In-house Dr Laura E McCoy, clone ID 03-018 ³)	69 [62-76]	1.6×10^5 [1.6-1.7]	5.8×10^{-3} [5.0-6.7]

Supplementary Information Table 1. Summary of capture antibody-antigen kinetic parameters including equilibrium dissociation constant, K_D , association rate, k_{on} , and dissociation rate, k_{off} obtained using BLI

Reviewer 1 Comment 6. 'In line 199: Authors mention that to reduce antibody concentration on TL they use strips with pre-printed streptavidin. In which percentage can they reduce the antibody concentration in comparison to non-streptavidin-printed strips? Could the authors add a reference supporting this?'

Response: We used streptavidin-printed test lines and a capture biotinylated antibody concentration of 7.7 μ g per 100 strips compared to our in-house antibody printed test strips at 1 mg/mL giving 50 μ g per 100 strips, a 84.6% reduction in antibody consumption.

Protocols from LFA manufacturer generally recommend printing capture antibodies at concentration of \sim 1 mg/mL to ensure effective capture and signal generation^{5,6}. For comparison, Grant *et al.* (2021)⁷, reported an assay with biotinylated antibody MM08 (Sino biological, the same detection antibody used in this work) sprayed on the conjugate pad in their integrated lateral flow assay. Their method sprayed the biotinylated antibody at 4

$\mu\text{L}/\text{cm}$ of $75 \mu\text{g}/\text{mL}$, roughly ~ 3.3 -fold less antibody than the standard used for test line printing at $1 \mu\text{L}/\text{cm}$ of $1 \text{ mg}/\text{mL}$.

This reduction also translates to a further reduction in total antibody consumption due to differences in application methods of direct antibody test line printing versus the solution-based biotinylated antibody and streptavidin test line method. Calculations for the solution method following methods described in the paper ($1 \mu\text{L}$ of $77 \mu\text{g}/\text{mL}$ biotinylated AbR001 per sample) are compared to the amount of antibody at $1 \text{ mg}/\text{mL}$ required for test line printing for a standard card (30cm nitrocellulose laminated) to produce 3mm width strips (Response Table 1). This gives a 6.5-fold reduction in antibody required per test. As antibodies are one of the main costs of lateral flow tests, this is an important reduction. The methods for in-house antibody-printed test strips have been added to Supplementary Information Section 1c.

	# of strips per 30cm card	Antibody Concentration (mg/mL)	Volume Ab required (μL)	Amount of Ab per 100 strips (μg)
Ab printed test line	100*	1	$50 \mu\text{L}$ *	50
Ab in solution	100	0.077	100	7.7
Ab consumption % difference				84.6%

Response Table 1: Comparison of quantities of antibody required for antibody vs streptavidin-printed strips. *denotes calculations that vary based on the manufacturing process. The yield on number of strips per card is fewer, on average, to account for discarded strips due to strip cutting variability and/or test line printing inconsistencies. The antibody volume required for test line printing includes a minimum of $20 \mu\text{L}$ required for dead volume of the Biodot AD1520 system

These costs are applicable to R&D scale manufacturing labs that manufacture test lines based on standard laminated cards. A reduction in antibody waste due to dead volume or discarded strips can be achieved in large scale reel-to-reel test line printing as used for commercially available PSA strips manufactured by Global Access Diagnostics.

Amendments to manuscript:

Main Text (lines 208-211):

'Antibody-printed test lines require high volumes and a large amount of excess antibody therefore we used biotinylated AbR001 (AbR001-b) in solution with polystreptavidin-printed test line to reduce antibody consumption by 84.6% and therefore cost.'

Supplementary Information Section 1c (Lines 54-61):

b) Lateral flow test line printing

CN95 nitrocellulose membrane at 2.5cm width (Sartorius) was laminated onto backing card (Kenosha) with 2.2cm absorbent pad (Millipore) and cut into 15mm card segments for printing (Biodot AD1520 dispenser). Antibodies were diluted from stock concentration to $1 \text{ mg}/\text{mL}$ in 1X PBS for printing at $1 \mu\text{L}/\text{cm}$, with test line printed 1cm from the bottom of the nitrocellulose membrane. Following printing, cards were placed in the oven to dry for 2hrs at 37°C . Strips were cut with strip cutter (ZQ2002, Shanghai Kinbio Tech. Co., Ltd.) at 3mm width. Strips were stored in a desiccated bag until use.

Reviewer 1 Comment 7. 'In extended data figure 2a not all error bars are well-visible.'

Response: We confirm that Extended Data Figure 2a does include small errors bars. To make the error bars more visible, we have now updated the figure with smaller markers. Error bars that are not visible represent a standard deviation < 0.05 , demonstrating minimal variability in test replicates. Error bars have been added to the SNR plot in Extended Data Figure 2b.

Amendments to manuscript:

Extended data Fig. 2a (Line 270): We have reduced the marker size to allow better visibility of some error bars. The caption for part (a) was updated to describe error and linear regression R-squared values were added to the caption. Error bars for the SNR were added to the plot in Extended Data Fig. 2b.

Extended Data Fig. 2: (a) FND concentration sweep from 6.6 fM to 42 pM tested on a negative sample ($n=3$) and positive sample (500pg/mL recombinant nucleocapsid protein $n=3$). Data points show mean and error bars show standard deviation (error bars not shown represent $s.d < 0.05$ units). Plots are fitted with linear regressions in Graphpad Prism; negatives (orange) R -squared=0.9545 and positives (blue) R -squared=0.9978. (b) SNR found by dividing the fitted linear regressions in (a), resulting in a constant value of ~ 20 . Dashed line represents s.d. error.

Reviewer 1 Comment 8. 'Why authors did not consider adding a control line on FND-LFTs? Please explain.'

Response: In this study we prioritized flexibility in test design and focused on analytical performance characterization and clinical validation. We used commercial single polystreptavidin test line (Global Access Diagnostics) half-strip solution-based assay format which offered adaptability to various assay design formats (e.g., direct binding, sandwich assays) and a wide range of antibodies while also reducing reagent cost. We implemented

negative and positive sample controls (n=3 per batch) to ensure reagent stability and test functionality, serving a similar purpose to a control line. The control line verifies that the strip has run correctly to the end – in this study of analytical performance (rather than a field study), the consistency between samples and compared to the positive controls verifies the strips have been run correctly. Future development into a product would incorporate a control line and cassette, but this is beyond scope of this paper.

Reviewer 1 Comment 9. ‘What about using a conjugate pad for the FNDs? May you explain?’

Response: Following the response above, this assay format allowed flexibility in the research phase of this important study. We plan to explore the use of a conjugate pad in future work. Surface functionalisation is similar to other particle systems, so we expect standard approaches will be applicable. Work is also on-going in the group to optimise and validate lyophilisation of reagents for a one-pot approach, and has been demonstrated in our group with a CRISPR-based real-time fluorescence assay in Cherkaoui *et al.* (2023)⁸. This approach is being explored for use with fluorescent nanodiamond based assays.

Assay kinetics are functionally similar with/without a conjugate pad: we show in Reviewer 1 Comment 11 (below) that the majority of particle binding happens at the test line to already-immobilised antigen. Additionally, we show in Reviewer 3 Comment 1b that the mixing time does not significantly affect SNR. Together these results support our working hypothesis that the sensitivity will be the largely similar when we progress to a conjugate pad.

Reviewer 1 Comment 10. ‘What is the purpose of adding 12.5mM MgCl₂ to PBS buffer during SEM characterization?’

Response: The addition of 12.5mM MgCl₂ to PBS buffer during SEM sample preparation serves to enhance particle adhesion to the surface. The divalent Mg²⁺ cations help bridging the substrate and particle surfaces, promoting better adhesion of the nanodiamonds to the substrate, thus achieving clean and clear scan images. This is a common technique used in SEM sample preparation to improve the distribution and immobilisation of nanoparticles.

Amendments to manuscript:

Methods (Lines 437-438): A line has been added to the SEM methods section to clarify the use of MgCl₂

Methods (Lines 437-438):

‘The addition of 12.5mM MgCl₂ improves particle adhesion to the substrate surface.’

Reviewer 1 Comment 11. ‘In the methods section, on FND lateral flow testing, would not have been better to immobilize the secondary biotinylated antibody on the streptavidin test line, instead of mixing it with the running buffer? Are the KD of the antibodies not enough to perform the recognition on the test line?’

Response: We thank the Reviewer for this suggestion. The chosen approach to mix the secondary antibody in solution primarily aimed to reduce our reagent cost (as described in

our costing table above) with antibody printed test lines and compatibility with the commercially available polystreptavidin test strips during our research.

To further investigate the effect of this assay design, we constructed a computational model of the assay considering mass transport and binding kinetics. The model is based on Gasperino et al.⁹ We model over time the concentrations of analyte, [A], antibody-particle conjugate, [P], biotinylated secondary antibody, [B], and streptavidin receptor, [R] (immobilized at the test line), and the various complexes of these components. We used measured binding rates of the antibodies (Supplementary Information Table 1), and the experimental parameters used in the real assay. Response Figure 3 shows this simulation for an analyte concentration of 41 fM, a low positive test. The new modelling demonstrates three key points in response to the Reviewers' comments 9 and 11:

1. The majority of analyte binding still takes place at the test line to antibodies that have already bound to the streptavidin ($[RB] + [A] \xrightarrow{k_{on1}} [RBA]$). This means the test is functionally similar to either printing or pre-immobilising the antibody. This is further demonstrated in Response Figure 4, showing the proportion of total analyte binding that takes place at the test line (as opposed to during running). This is high for most of the running time of the test (lower near the beginning as the amount of antibody at the test line is increasing from zero).
2. The smaller part of analyte binding that takes place before the test line is predominantly binding to the biotinylated antibody $[B] + [A] \xrightarrow{k_{on1}} [BA]$, which binds, in turn to receptors at the test line to form the $[RBA]$ complex.
3. The vast majority of particle binding happens at the test line to already-immobilised analyte, particularly close to the LoD ($[RBA] + [P] \xrightarrow{k_{on2}} [RBAP]$). This mean conjugate pad and mixing time are largely unimportant to sensitivity.

Bringing this together, Response Figure 5 below shows the total test line complex, comparing this assay format (RBAP) and a conventional sandwich assay (RAP). We used the same protein loading per membrane surface area (either streptavidin or antibody). This shows similar total binding for the two assays, conveying similar LoDs. Furthermore, our in-house studies of antibody printed test strips (Extended Data Fig. 1d) demonstrate that the antibody K_D is sufficient to capture the complex directly at the test line (without the biotin-avidin interaction) and is a feasible method to reach comparable sensitivity.

Response Figure 3: A simulation of the LFA for an analyte concentration, $[A]$, of 41 fM, a low positive test. The concentrations of LFA components, and complexes, are plotted over time (colour) and distance along the strip (x axis). $[P]$ is the antibody-particle conjugate, $[B]$ the biotinylated secondary antibody, and $[R]$ the streptavidin receptor immobilized at the test line.

Response Figure 4: A simulation of the analyte binding in an LFA. Plotted is the proportion of total analyte binding that takes place at the test line (TL) (as opposed to during running). This is independent of analyte concentration – the proportion is the same for each concentration. The early spike ($t = 61$ sec) is the initial wetting of the strip.

Response Figure 5: Modelled total test line complex, comparing this assay format (RBAP) and a conventional sandwich assay (RAP). This model shows similar total binding for the two assays, conveying similar sensitivity.

Reviewer #2 (Remarks to the Author):

The manuscript “Quantum-enhanced nanodiamond rapid test advances early SARS-CoV-2 antigen detection in clinical diagnostics” presents the clinical application of a quantum-enhanced nanodiamond rapid test for early SARS-CoV-2 antigen detection. This study builds upon their previous work, “Miller, Benjamin S., et al. Spin-enhanced nanodiamond biosensing for ultrasensitive diagnostics. *Nature* 587.7835 (2020): 588-593.”, which introduced the concept of nanodiamond-based lateral flow tests (LFTs) for rapid early-stage disease detection. While the clinical validation of this method is of interest, the manuscript primarily focuses on methodological validation rather than presenting a significant conceptual or technological advancement. As a result, its novelty is marginal in relation to the standards of *Nature Communications*. Additionally, several other concerns need to be addressed. Therefore, in its current form, the manuscript does not appear suitable for publication in *Nature Communications*.

Response: We thank the Reviewer for their assessment of our manuscript and for identifying potential interest in the clinical validation presented in this work. We believe this work presents important progress in the field of nanodiamond rapid diagnostics, particularly for early detection of infectious diseases, addressing two critical gaps in the literature: (i) development and first large-scale evaluation with *unprocessed* real clinical samples with high accuracy and (ii) quantifying the clinical impact of early detection – the first analysis of this kind on nanodiamond LFTs.

- (i) There are significant hurdles transitioning a diagnostic technology from model systems to complex and variable clinical samples. Clinical samples (e.g., nasopharyngeal swabs) contain a diverse array of biomolecules, including mucins, proteins, lipids, and cellular debris, which can significantly impact the performance of highly sensitive nanoparticle-based assays¹⁰. These components can lead to non-specific binding (NSB)¹¹, aggregation of nanoparticles, and altered analyte kinetics, all of which can severely compromise sensitivity and specificity. This point is often obscured because many studies do not thoroughly evaluate performance in clinically relevant/real patient samples.

Nanodiamond diagnostics are to-date mostly demonstrated in model systems or spiked sample solutions, leaving their real-world clinical suitability unclear. Studies that do evaluate clinical samples frequently report a significant drop-off in performance compared to model conditions¹². We have also noted in Main Text Lines 70-78, the significant difficulties other groups have faced when moving assay systems to work with clinical samples, highlighted by Feuerstein *et al.*¹³ who saw a 100 to 200-fold reduction in performance moving to serum samples in their FND-based LFT for Ebola glycoprotein detection. This highlights a critical gap in the field's understanding of how these advanced nanomaterial-based diagnostics perform under complex biological conditions.

Our work demonstrates a quantum-enhanced nanodiamond LFT that maintains favourable kinetics with minimal non-specific binding in unprocessed clinical samples, a crucial step towards translation of the FND platform into a robust and accurate clinical diagnostic. We use qPCR and LFTs to make a direct comparison between spiked inactivated virus and clinical samples, demonstrating only a ~29% decrease in the limit of detection from model inactivated virus (LoD= 3.4×10^4 copies/mL [95%CI: 2.8×10^4 -

8.6 x 10⁴) sample to clinical samples (LoD= 5.6 x 10⁴ copies/mL [95% credible interval: 4.2 x 10⁴ to 7.5 x 10⁴]). This comparison provides crucial and previously unavailable data on the real-world potential of the nanodiamond platform.

- (ii) We then take a step further to real world utility by quantifying the clinical impact of this assay using real clinical data to directly translate the diagnostic sensitivity to direct advantage in real-world applications. We showed an average diagnostic advantage of 2.0 days earlier than conventional gold nanoparticle LFTs with identical antibodies (diagnosis is possible just 0.6 days after RT-PCR). The improved sensitivity would have detected ~69,000 more patients in a single day at the peak of Omicron.

Together these results – demonstrating clinical suitability and clinical impact – are essential steps towards the real-world adoption and impact of this technology, and the field of quantum sensing. This study is novel and impactful both in methodology and results.

Reviewer 2 Comment 1. ‘The control line is missing in your nanodiamond-based LFTs. In LFT, the control line is essential for ensuring the validity and reliability of test results. Without a control line, it is impossible to determine whether the test functioned correctly or failed due to factors such as insufficient sample volume, reagent degradation, or improper sample application. Additionally, the absence of a control line increases the risk of false positives, which may arise from non-specific binding, background noise, or improper flow, falsely indicating the presence of the target analyte.’

Response Reviewer 2 Comment 1:

Also see response to Reviewer 1 Comment 8.

In this study we implemented negative and positive sample controls (n=3 per batch) to ensure reagent stability and test functionality, serving a similar purpose to a control line. The control line verifies that the strip has run correctly to the end – in this study of analytical performance (rather than a field study), the consistency between samples and compared to the positive controls verifies the strips have been run correctly. We chose to prioritize flexibility in test design and focused on analytical performance, characterization and clinical validation over final device design. We utilised the commercial single polystreptavidin test line (Global Access Diagnostics) half-strip solution-based assay format which offered adaptability to various assay design formats (e.g., direct binding, sandwich assays) and a wide range of antibodies while also reducing reagent cost during development. Future development into a product would incorporate a control line and cassette, but this is beyond scope.

Reviewer 2 Comment 2a. ‘Regarding limit of detection (LOD), existing studies on LFT-based SARS-CoV-2 rapid detection have already demonstrated highly sensitive performance. For instance, an LFT-based antigen detection method has reported a detection limit as low as 0.5 pg/mL (ACS Applied Materials & Interfaces, 2021, 13(34): 40342–40353). Given this benchmark, what specific advantages does your approach offer in terms of sensitivity, reliability, or clinical applicability?’

Response: We thank the Reviewer for raising the crucial point regarding the LOD in the context of existing LFT-based SARS-CoV-2 rapid detection methods. We have compared our

work with referenced study, which reports an LOD of 0.5 pg/mL for recombinant antigen. We believe there are five advantages of our work:

1. Our LoD is for real clinical samples - nasal and nasopharyngeal swab samples from individuals at UCLH (including travellers, healthcare workers, or other patients)

Wang et al. use model samples - spiked samples (spiked with pseudovirus) to test their assay and quantify LoD in copies/mL (Wang *et al.*¹⁴ Figure 6). We use real, unprocessed clinical samples - acquired from individuals UCL Hospital containing real virus to quantify LoD in copies/mL, benchmarking to qPCR (Thomas DeCruz et al. Figure 3). While some matrix effects can be demonstrated using spiked samples in a relevant matrix, the population variance in real unprocessed clinical samples is an important consideration that can't be evaluated in spiked samples only. As we emphasize in our manuscript (lines 361-368), the performance of rapid antigen tests for SARS-CoV-2 has shown significant variability depending on the patient sample pool. This is directly attributable to the complexity of clinical samples including varying viral loads, varying concentrations of other biomolecules, and the efficiency of viral lysis for antigen release. Spiking consistent sample matrices often leads to an overestimation of the assay's true performance in a clinical setting. We have shown excellent sensitivity and specificity in clinical samples, and test against real negative controls sampled from the general population, not pre-COVID or mock samples which adds to the robustness of this work. Moreover, the model spiked pseudovirus tested in Wang *et al.* only presents the spike protein and therefore cannot attest to the efficiency of viral lysis required for detection nucleocapsid protein which is better demonstrated in our model inactivated whole virus sample.

2. We use a simple assay with no magnetic bead preconcentration

The 0.5 pg/mL LoD to recombinant antigen reported in the Wang et al. is achieved through a more complex assay workflow that includes an additional magnetic bead concentration step. They report 1 pg/mL without this additional step. Our recombinant antigen results using the quantum-enhanced nanodiamond LFT demonstrate an LoD of 0.67 pg/mL [95% CI: 0.31-1.4] without the need for any additional pre-concentration steps. This format is more practical, cost-effective, and amenable to point-of-care settings.

3. We use more robust LoD calculations and ROC analysis

Wang et al. calculate their LoD a threshold of the mean + 3 s.d.s of the negative controls. This method only accounts for variability in the negative controls, making an implicit assumption that variance is similar in positive samples. They also do not quote confidence intervals for their LoDs. Our method accounts for variance in negative and positive samples separately, and calculates confidence intervals for the LoD. In addition, our extension to clinical samples allows ROC analysis, and calculation and discussion on the clinical sensitivity, necessary for translation.

4. Simplified nanoparticle synthesis for scalability

Beyond performance metrics, the practicality and scalability of the diagnostic platform are critical for widespread clinical applicability. Our fluorescent nanodiamonds are synthesized using a well-established, low-cost process involving High-Pressure/High-Temperature diamond synthesis, milling, and thermal annealing. In contrast, the composite particles utilised in the Wang et al. require a more complex chemical synthesis route. Simpler particle synthesis offers advantages in cost-effectiveness, scalability, and accessibility, for a wider clinical impact.

5. Demonstrating clinical impact through population-level modelling

Finally, the clinical applicability of our test is further substantiated by the inclusion of population-level modelling. This analysis goes well beyond simple sensitivity and specificity metrics to explore the potential impact of our rapid test at scale, demonstrating its capacity for earlier detection and identification of a greater number of infected individuals within a population. This forward-looking analysis provides compelling evidence for the potential clinical utility and public health impact of our quantum-enhanced nanodiamond LFT.

Reference in manuscript:

Main Text (Lines 239-243): Wang *et al.* is referenced (ref #43 in main text) in sensitivity comparison to other reports in literature

‘The analytical sensitivity reached in this work demonstrates superior sensitivity of spin-enhanced LFTs compared to similar low-cost rapid LFTs reported in literature^{16,38,43-45} (Supplementary Information Table 3) and reaches LoDs comparable to high-throughput, microarray platforms such as Quanterix’s Simoa.’

Supplementary Information Section 3c (Lines 99-101): Reported sensitivity of Wang *et al.* in Supplementary Information Table 3, Supplementary Information ref #10

a) Comparison of assay sensitivities reported in literature

Reference	Sensor type	LoD	Clinical evaluation high viral load (Ct≤25 or 10 ⁶ copies/mL) sensitivity
Chen et al. (2022) ⁷	Multiplex fluorescent quantum dot LFT	10 pg/mL	No clinical samples tested
Grant et al. (2021) ⁸ Bachman et al. (2021) ⁹	Latex bead LFT	200 TCID ₅₀ /mL	92% (n=72)
Wang et al. (2021) ¹⁰	Magnetic quantum dot LFT	1 pg/mL (direct mode)	No clinical samples tested
Wei-Wen Hsiao et al. (2022) ¹¹	Fluorescent nanodiamond LFT (magnetic modulation)	1.94 ng/mL	No clinical samples tested
Gupta et al. (2023) ¹²	Fluorescent gold nanorod LFT	212 pg/mL	97.5% (n=40)
This work	Fluorescent nanodiamond LFT (MW modulation)	0.67 pg/mL	100% (n=30)

Supplementary Information Table 3: Summary of SARS-CoV-2 Ag-LFTs targeting nucleocapsid protein reported in literature with their assay limit of detection.

Reviewer 2 Comment 2b. ‘Additionally, your detection system involves a high-speed camera and devices for microwave source, which increase the cost of detection. The procedure also requires professional operators, which conflicts with the principles of LFTs, designed to be portable, rapid, and cost-effective.’

Response: We agree these are crucial considerations so would like to clarify the cost and operational requirements of our detection system. Our method does not require a high-speed camera – we use 50 fps, but frame rates down to 8 fps are suitable (or lower by lowering the modulation frequency). This is achievable with very low-cost cameras. Whilst microwave components may increase cost at the research scale, the components are miniaturised, and low-cost when manufactured at scale (e.g. for smartphone communications). Our group is currently developing a field-deployable smartphone connected system that will be presented in an imminent publication, but is beyond the scope of this publication. We have included the costing of this device below in Response Table 2 and added discussion to Supplementary Information Section 9c. At present our standalone device is ~£912 at research scale buying individual components, a smartphone (the latter is the largest single cost) and a power bank.

Device	Description	Cost (£)
FND prototype smartphone reader	Resonator	6
	30mW laser	93
	Lens & filters	221
	Cables & 3D printed material	25
	Voltage controlled oscillator & chip	65
	Amplifier	137
	Power bank	20
	Smartphone (LG-G7)	345
	Total	912

Response Table 2: Estimated costing of FND portable reader with smartphone connectivity in development.

The need for a reader device (as in Wang et al.) to enhance the sensitivity and reliability of LFTs is increasingly recognized in the field, not just as necessary, but desirable for data capture and connectivity to health data systems. This is particularly relevant for early-stage disease detection where analyte concentrations are low. The REASSURED criteria list connectivity (using a reader) as a feature of an ideal diagnostic test¹⁵. They specifically highlight the use of mobile phone-based readers for LFTs as crucial for data analysis, transmission and collection for individual and population surveillance, facilitating centralised decision making, patient retention and faster access to treatment. The 100 Days Mission for Pandemic Preparedness also highlights the vital role of connected lateral flow tests¹⁶.

Amendments to manuscript:

Supplementary Information (Lines 295-305): Additional section 9c addresses feasibility of developing a prototype reader for the FND system

Supplementary Information (Lines 295-305):

c) Estimated costs of portable fluorescent reader

The feasibility of a portable fluorescent reader with modulation capabilities is a key component to the translation of the FND platform to a point-of-care setting. A portable, smartphone-based read-out system meets point-of-care criteria using cost effective 3D-printed housing, miniaturised electronics, and small resonator components allow for

integration into a portable device that can be deployed in the field and at the point-of-care including doctors' offices and pharmacies. A costing summary of a prototype FND device is shown in Supplementary Information Table 6.

Response Table 2 added to Supplementary Information Section 9c as Supplementary Information Table 6.

Reviewer 2 Comment 3a. 'Noise interference presents a significant challenge in LET-based detection, directly affecting the limit of detection (LOD). In Extended Data Fig. 3(c), in addition to the color change in the C and T lines, a faint red background is visible. This is likely due to the partial retention of gold nanoparticles on the NC membrane during the flow process. A similar issue may arise with FNDs, particularly given the relatively large 600 nm nanodiamonds used in this study. To mitigate this, further optimization of the test strip and improvements in the signal-to-noise ratio should be considered.'

Response: We thank the Reviewer for highlighting test strip noise and the relatively large size of the 600nm FNDs. We have performed optimisation studies, reported in Miller et al., to maximise the SNR including selection of polyglycerol-functionalised FNDs. Furthermore, both in this work (Extended Data Fig.2) and Miller et al., buffers were optimised to reduce the non-specific binding. To clarify, we have added additional buffer optimisation data to Extended Data Fig. 2c,d that further supports robustness of assay optimisation. Methods detailing the optimised running buffer formulation and gold nanoparticle antibody conjugation have been added to the Supplementary Information Section 1.

In order to evaluate whether significant quantities of FNDs are retained by the membrane, we performed pixel-wise lock-in analysis at 10X magnification and 50 frames per second on a positive (Ct=25) and negative clinical sample, where pixel brightness represents the specific nanodiamond lock-in signal. We can use this to visualise *where* nanodiamonds are in the membrane. This analysis demonstrated minimal discernible residual FNDs in the membrane surrounding the test line, supporting our conclusion that unwashed FNDs from the membrane did not significantly affect the signal-to-noise ratio in our FND assay (Response Figure 6).

Response Figure 6: (a) Optical image of FND test line from a positive clinical swab sample tested on the FND Ag-LFT taken with a 10X magnification lens and 20ms exposure (50fps) on the microscope set-up, showing a

wider area of interest around the test line. For comparison, the red box shows the region imaged for the analysis in this work using a 20X magnification lens and 20ms exposure. A pixel-wise lock-in analysis was performed where brightness corresponds to higher lock-in signals or detectable FNDs. **(b)** 10X test line image of a negative SARS-CoV-2 clinical swab sample.

Amendments to manuscript:

Extended Data Fig. 2 (Lines 570-587): Evaluation of buffer pH and NaCl concentration was added to the bar graph with buffer detergent (Triton X-100 and IGEPAL CA-630) optimisation studies shown in part c. Part D, the data was expanded to include additional buffer concentrations of Tween-20 and Casein to further support robustness of assay buffer optimisation.

Extended Data Fig. 2: FND Ag-LFT optimisation

Extended Data Fig. 3: **(a)** FND concentration sweep from 6.6 fM to 42 pM tested on a negative sample ($n=3$) and positive sample (500pg/mL recombinant nucleocapsid protein $n=3$). Data points show mean and error bars show s.d. error. Plots are fitted with linear regressions in Graphpad Prism; negatives (orange) R -squared=0.9545 and positives (blue) R -squared=0.9978. **(b)** SNR found by dividing the fitted linear regressions in (a), resulting in a constant value of ~ 20 . Dashed line represents s.d. error. **(c)** SNR testing different buffer components for viral lysis using whole inactivated virus sample (7.9×10^3 TCID₅₀/mL) as positive sample ($n=3$). Buffer pH8 was selected and tested with addition of NaCl, where 100 mM NaCl only showed significant difference from 50 mM (p -value=0.0323, one-way ANOVA, $F=4.70$, $n=8$, post-hoc Tukeys multiple comparisons). IGEPAL CA-630 showed better SNR at both concentrations although not a statistically significant difference determined by one-way ANOVA p -value=0.21, $F=2.1$, $n=9$, post-hoc Tuckey's multiple comparisons. Error bars represent standard error of the mean. Data points represent the SNR of individual test replicates by mean baseline signal. Buffer was tested independent of additional protein blockers (shown in part d) resulting in overall poorer contrast **(d)** Varying concentrations of Tween20 and Casein in the buffer to determine optimal concentration of blocking components.

Heat map shows mean SNR of a negative (n=3) and positive (500pg/mL recombinant nucleocapsid protein n=3) where the addition of 0.8% casein and 0.05% Tween20 showed the best SNR of 23.

Supplementary Information Section 1b (Lines 39-53):

b) Running buffer formulation

A clean beaker was placed on a magnetic stir plate with a magnetic bar for continuous stirring. Formulation was prepared at 5X concentration of IGEPAL CA-630, Casein, and Tween20 in 50mM Tris-HCl + 100mM NaCl base buffer. 50mM TRIS-HCl solution was made up to pH 8 mixing appropriate amounts of Trizma Hydrochloride (MW: 157.60 g/mol) + Trizma Base (MW: 121.14 g/mol) (mg) in DI water, continuously stirring until fully dissolved. NaCl (Sigma) was added at 100mM in final solution volume, continuously stirring until fully dissolved. The 50mM Tris-HCL + 100mM NaCl buffer was split into two batches, one stored at 2-8°C until further use, and the second batch was used to prepare the 5X buffer. While continuously stirring, 10% w/v IGEPAL CA-630 was slowly added to the mixture, followed by 0.25% w/v Tween20. Next, 4% w/v Casein (Hammarstein bovine, Sigma Cat#E0789) was added and the temperature was raised to 40°C monitored by a temperature probe, covered, and continuously stirred for 24hrs. Following, the buffer was aliquoted and frozen at -20°C. The 5X buffer was thawed and diluted 1:5 in base buffer (50mM Tris-HCL, pH8 +100mM NaCl) to make a 1X solution and stored at 4°C for lateral flow testing.

Supplementary Information Section 1d (Lines 62-71):

d) Gold nanoparticle (AuNP) functionalisation

The parameters for antibody physisorption were determined using 40 nm citrate AuNPs (Nanocomposix) at OD1 and swept across antibody concentrations from 8.7 to 70 µg/mL, in buffers ranging from pH 7.7 to 9.0. A salt stress test was applied and particle absorption spectra was measured using plate reader (SpectraMax i3, Molecular Devices) to check for aggregation and absorption properties. A 1mL aliquot of AuNPs was mixed with 200µL of AbMM08 at 35µg/mL in borate buffer pH 8. This mixture was incubated at room temperature for 1hr at 650RPM (Thermoshaker). Blocking was performed by adding 100µL of 1mg/mL BSA in H₂O and left to shake for 30mins. The mixture was washed by centrifuge (3 washes at 10,000 rcf for 10mins) and resuspended in 5% BSA + 0.05% Tween20 in PBS.

Reviewer 2 Comment 3b. 'Additionally, in Extended Data Fig. 2(b), noticeable inconsistencies in brightness and contrast are observed across the six test strip images. If these variations are due to differences in photography conditions, capturing all six samples in a single shot could help standardize image quality and improve comparability.'

Response: We thank the Reviewer for noting this oversight. The strips were photographed in a light-controlled box, however the dimensions restrict capturing n=39 test strips in a single shot and other factors such as curvature of the lateral flow strip can lead to variation in lighting conditions produced by shadows and reflection. The test line analysis involves normalisation of the image colour profile particular to the selected test line region of interest and is therefore insensitive to changes in background, differing from a visual read-out. To justify this claim, we have normalised the images, then refitted the limit of detection to show the lighting conditions did not significantly affect the limit of detection (Response Figure 7a,b, t-test p-value=0.563, T=0.058, DF=70.6). We have also shown the mean test line

intensity profile showing the variation in brightness of the low concentration test strips does not affect the processed intensity profiles (Response Figure 7c).

Response Figure 7: (a) Comparison of limit of detection from the raw (original) images and normalised images showing lighting conditions and noise didn't have a significant impact on achievable LoD (t -test p -value=0.563, $T=0.058$, $DF=70.6$). (b) Images of original pre-processed images (top) and processed images with colour profile normalisation (c) Mean test line intensity profiles ($n=3$ test replicates) of imaged test concentrations.

Amendments to manuscript:

Extended Data Fig. 3b (Lines 588-600): Replaced AuNP test line images with pre-processed images with normalised brightness.

Extended Data Fig. 3: Assay test line analysis across FND LFT, in-house gold LFT and a commercial test

Extended Data Fig. 4: (a) LoD with recombinant nucleocapsid protein on FND assay. Images of FND Ag-LFT test lines from selected concentrations of serial dilution with corresponding fluorescent intensity time series plots (bottom) showing detectable periodic signal after test line is no longer visible in the images. Arrow shows the approximate LoD calculated from the exponential fit (Fig. 2b). (b) Select concentrations of strips from serial dilution of recombinant nucleocapsid protein. Samples were ran in triplicate for each sample concentration. AuNP assay strips were ran for 15mins and imaged by camera in a light-controlled box. Images of the test line were then analysed through MATLAB using a pixel-wise line intensity plot. Arrow points to the calculated LoD from exponential fitting curve. (c) Images of FlowFlex SARS-CoV-2 Ag assay tested with selected concentrations of gamma irradiated wild-type viral isolate following the product insert protocol. Arrow indicates approximate concentration of lowest detectable test line at $\sim 10^3$ TCID₅₀/mL.

Reviewer #3 (Remarks to the Author):

In the manuscript titled “Quantum-enhanced nanodiamond rapid test advances early SARS-CoV-2 antigen detection in clinical 3 diagnostics”, DeCruz et al. demonstrated a spin-enhanced nanodiamond LFT with sub-pg/mL analytical sensitivity. Moreover, they demonstrated that this analytical sensitivity translated directly to high clinical sensitivity for SARS-CoV-2 with 100% specificity. This improved analytical sensitivity also improved time-to-detection relative to traditional gold nanoparticle based LFTs.

The highly-sensitive LFT described in this manuscript provides a significant advancement to the field of rapid diagnostic tests for Covid-19 and potentially other LFAs for settings in which readers are appropriate. The authors utilized appropriate and rigorous statistical methods, e.g. Dunnett’s test and ANOVA when examining cross-reactivity, and ANOVA when comparing antibody performance. The data is high-quality and presented clearly. Analytical tests were conducted in a relevant matrix (VTM). The samples for the clinical study were also chosen appropriately – the authors utilized symptomatic patients or those returning from travel instead of utilizing healthy controls for their negatives. Selecting additional samples near the LOD to more accurately assess the limit-of-detection was also an appropriate choice. The high specificity is due to choosing a threshold based on an ROC curve; however, this is not a criticism but rather an intrinsic benefit of utilizing this technology instead of a traditional visually read LFT.

The authors correctly identified limitations of their study; for instance, they advise caution in analyzing data that is based on CT values from two different studies (e.g. figure 4C and lines 330-332). They additionally note difficulties in establishing a representative clinical pool and correctly note their study population may lead to underestimating clinical sensitivity while other studies may tend to overestimate sensitivity. The inclusion of their own gold LFT performance helps to elucidate the effect of the FND approach within the same clinical pool. While the research is ultimately very promising, the utility will be limited until a point-of-care compatible reader is realized in a way that allows use in settings that benefit from LFA use vs other existing sensitive covid assays. In the interim, traditional LFTs will continue to be utilized despite their sensitivity limitations.

Response: We thank the Reviewer for stating that our *‘highly-sensitive LFT described in this manuscript provides a significant advancement to the field of rapid diagnostic tests for Covid-19 and potentially other LFAs for settings in which readers are appropriate’* and note the *‘rigorous statistical methods ... data is high-quality and presented clearly.’*

We agree with the Reviewer regarding the importance of a point-of-care reader to enable utility of the FND platform at the point-of-care. Our group is currently developing a field-deployable smartphone connected system that will be presented in an imminent publication, but is beyond the scope of this publication. Please see Reviewer 2 comment 2b response.

Reviewer 3 Comment 1a:

‘Major issues:

Methods: The decision to utilize LFAs in a format with all wet reagents needs to be included in the discussion. My understanding from the method is that wet biotinylated antibody, sample and FNDs are placed in a 96-well plate and allowed to bind for 10 minutes prior to running them on an LFT. In a traditional LFT, the sample would rehydrate the conjugate and bind during flow. Do the authors have references for this being an acceptable approximation?’

Response: We thank the Reviewer and agree the decision to use the lateral flow half-strip assay with reagents in solution is important to address. We have now performed additional experiments to investigate the 10-minute binding step detailed in Supplementary Information Section 9a. We tested lateral flow tests at 1-minute intervals from 0-10 minutes, where time 0 represents no wait time i.e. immediately adding the strip to the wells containing FNDs and sample, and 10 mins is equivalent to the protocol followed in this paper. A low positive analyte concentration of 31 pg/mL (close to the detection limit) was used to evaluate the SNR where binding interactions are slower and therefore more likely to be impacted by binding time. We found no significant difference between mean SNRs across the time range (Response Figure 8, one-way ANOVA, Tukey’s multiple comparisons, p-value=0.1479, F= 1.68, DF=32).

Response Figure 8: The SNR of negative (n=3) and low positive (31 pg/mL, n=3) tested at 1 min time intervals where at time 0, the lateral flow strips were ran immediately after addition of capture/detection reagents. Data points represent mean SNR with error bars representing the standard error from the mean, showing no significant difference between mean SNRs across the time interval (one-way ANOVA, Tukey’s multiple comparisons, p-value=0.1479, F= 1.68, DF=32) and a gradient not significantly different from zero using a linear regression fit (95%CI: -0.0089 to 0.67).

This result is supported by modelling work, based on Gasperino et al.⁹, outlined in Response to Reviewer 1 Comment 11. This shows that the majority of particle binding happens at the test line between non-complexed particles, and already-immobilised analyte. This means the wait time and flow time are less important. This is shown in Response Figure 9, below:

Response Figure 9: A simulation of particle (FND) binding in an LFA over time for a range of concentrations used in this work. Plotted is the proportion of total FND binding that takes place at the test line (TL) to already-immobilised analyte (as opposed to during running). The early spike ($t = 61$ sec) is the initial wetting of the strip.

Amendments to manuscript:

Main Text (Lines 379-383): A paragraph has been added to the discussion to address the additional experiments and future work to demonstrate full device integration.

Main Text (Lines 379-383):

‘Future work involves moving towards a cassette-based lateral flow test platform with integrated reagents. Initial experiments described in Supplementary Information section 9 demonstrate adequate reaction kinetics required for an integrated device where capture and detection of reagents typically occurs during rehydration of the conjugate and flow up the membrane (Supplementary information Fig. 5).’

Supplementary Information (Lines 221-260): Section 9a was added detailing the experiment performed to demonstrate adequate reaction kinetics for a full-integrated device.

Supplementary Information Section 9 (Lines 221-260):

9) Assessing performance retention transitioning to an integrated point-of-care platform

The assay presented has been designed for greater flexibility in design and optimisation throughout the initial stages of development in a lab-based setting. This section addresses the feasibility of retaining similar performance in a fully integrated lateral flow platform. These considerations include (i) adequate reaction kinetics for on-pad reagent delivery in cassette format (ii) immediate read-out of strip following 15min run time (without drying) for fast time-to-results and (iii) feasibility of cost-effective, portable fluorescence read-out device.

a) Reaction kinetics study: Demonstrating equivalent assay performance with direct strip application.

This study shows feasibility of eliminating the 10-minute incubation of reagents prior to adding the lateral flow strip, which more closely represents the binding interactions that occur in a cassette-based lateral flow format as complexes are formed and captured immediately during flow up the membrane. Strips were tested with a wait time from 0-10 mins at 1 min intervals, evaluating the impact on the SNR from a low positive (31 pg/mL) and negative sample ($n=3$) over time. A low analyte concentration (31 pg/mL) was chosen for this study to reflect a case where complex formation/binding is less favourable and therefore more likely to observe potential impact on performance than at high concentrations where binding time has diminished impact.

The assay was performed as follows: 5 μ L of the FNDs (26 fM) + 49 μ L of sample diluted in running buffer + 1 μ L of bAb01 (513nM) was added to wells in a 96-well plate and allowed to bind for 0-10 mins. Strip were added to the wells and run for 15 mins, followed by test line read-out of dry strips. Time 0 mins corresponds to application of the lateral flow immediately after adding reagents to the wells and 10 min wait time is representative of the protocol used throughout this work. No significant difference in SNR was observed across the time intervals tested (Supplementary Information Fig. 5), indicating the complex formation in solution does not impact effective capture at the test line. In future, eliminating the “binding time” step for this assay can reduce assay time to the standard \sim 15 min lateral flow test run time without impacting performance.

Response Figure 8 added to Supplementary Information Section 9a as Supplementary Information Fig. 5

Reviewer 3 Comment 1b:

‘Second, please include some discussion as to the decision to allow the strips to dry prior to reading them. Typically, LFTs will be read within 10 to 15 minutes while the nitrocellulose is still wet – not dry.’

Response: We agree this is a crucial consideration for LFT development. For this study, we ran many strips at once and wanted to be consistent, so we measured all strips dry. We have performed an additional experiment that evaluates the impact on lock-in values when read-out of a test strip is performed immediately after running (strip is still wet) detailed in Supplementary Information Section 9b. The lock-in measurements were taken at 1 min time intervals over 60 mins (Response Figure 10). The key potential confound for wet strips is the shift in the resonator’s resonant frequency due to changing dielectric constant of the test caused by strip wetting. There are two important effects:

1. We observed a small reduction in lock-in values in the first \sim 10-15 minutes of read-out, likely due to FNDs still washing up the strip, reducing signal. This effect competes with the drying, which is concurrently increasing signal. In the positive sample, a portion of these particles bind at the test line, rather than flowing past (as in the negative), giving a smaller dip. This happens in all LFTs, not just nanodiamond LFTs. This is a higher proportion of the final lock-in value for the negatives, as the signal is lower. The overall effect on SNR is an increase as the final particles run through the strip. It will have a minimal effect on sensitivity and can be mitigated with a differential readout (normalising to strip background).
2. Over the subsequent period of about 5-minutes, the signals rise in both positive and negatives tests as the strips dry. The overall effect on SNR is therefore zero. This can therefore be mitigated by several techniques:
 - a. Taking a ratiometric measurement of test line and background (analogous to test line and signal)
 - b. Taking a ratiometric measurement of test line and control line
 - c. Tuning the resonator to account for the changes in resonant frequency with wet strips (either in real-time or preset).

Response Figure 10: (a) The normalised FND lock-in signal of $n=3$ negatives (grey) and $n=2$ positives (blue) measured at 1-minute intervals over 60 minutes, shaded region showing standard deviation. The effect of wetting reduced lock-in signal of positive samples by $\sim 40\%$ of its maximum value at time 0, with a coefficient of variation (CV) = 18.3% across all time points. Minimal effect on the negatives was observed CV=11.2% across all time points (no error bars shown for error <0.1). Inset shows region from 0-15 minutes on a linear scale, where an initial decrease in lock-in signal is observed, likely due to FNDs washing away while the strip is still wet and solution is still moving across the strip. This is then counterbalanced by the strip drying, increasing signal again. In the positive sample, many of these particles bind to the test line rather than flowing past, giving a smaller dip. **(b)** The signal-to-noise ratio was calculated from the means in (a), with a minimum SNR of $8 (\pm 1.1)$ at time 0 and reaching a plateau ~ 11 mins with $\text{SNR}=13 (\pm 1.2)$. The SNR has variation of 11.2% across 60 mins.

We have added these results and accompanying discussion to the Main Text and Supplementary Information Section 9b.

Amendments to manuscript:

Main Text (Lines 383-396):

‘The lateral flow strip read-out is a key consideration for fluorescence-based assays. More specific to our FND-based modulated platform, we considered the dielectric effect from a wet lateral flow strip on the resonator and subsequent lock-in result to allow for immediate read-out. The effect of wetting of the lateral flow strip showed a $\sim 40\%$ reduction (small compared to the LOD uncertainty and comparable to the strip-to-strip coefficient of variation of 18.3%) in the lock-in signal with immediate read-out after LFT running compared to after 20 minutes (Supplementary Information Fig. 6). However, this effect can be accounted for by normalising to the test membrane, or control line, or using a tuneable resonator to mitigate shifts in resonant frequency as the strip dries (Supplementary Information section 9b). The small size of the resonator and high brightness of FNDs allow for this system to be integrated into a portable, cost-effective smartphone fluorescent reader with estimated production cost of our prototype device at $\sim \text{£}912$ (Supplementary Information section 9c, Supplementary Information Table 6).’

Supplementary Information Section 9b (Lines 262-293): Additional experimental section part 9b was added to demonstrate feasibility of immediate read-out of the strip.

Supplementary Information Section 9b (Lines 262-293):

b) Impact of lateral flow test strip drying on FND lock-in signal

This short study looked at the effect of reading wet lateral flow strips on the resonator-based read-out platform. This would allow for read-out of lateral flow strips immediately following running, reducing the total assay time to fit the point-of-care criteria. The key potential confound for wet strips is the shift in the resonator's resonant frequency (water has a high dielectric constant).

Test strips from a positive (n=2) and negative (n=3) sample were measured using lock-in analysis at 1-minute time intervals from 0-60 mins from the end of strip running. We observed a ~5 point reduction in lock-in values in the first ~10 minutes of read-out, likely due to FNDs still washing up the strip. This happens in all LFTs, not just nanodiamond LFTs. This is a higher proportion of the final lock-in value for the negatives, as the signal is lower, so there is a net effect on SNR: lower immediately after reading. It will have a minimal effect on sensitivity and can be mitigated with a differential readout (normalising to strip background).

Over the subsequent period of about 5-minutes, the signals rise in both positive and negatives tests as the strips dry. The overall effect on SNR is therefore zero. This can therefore be mitigated by taking ratiometric measurements of test line and background (analogous to test line and signal), or test line and control line. Finally, the resonator could be tuned to account for the changes in resonant frequency with wet strips (either in real-time or preset).

Response Figure 10 added to Supplementary Information Section 9b as Supplementary Information Fig. 6

Reviewer 3 Comment 1c:

'Finally, the impact of this device relies on making a low-cost portable reader. The authors state that is possible (line 237) but should add a line or two explaining the feasibility of this to the discussion.'

We have added an additional section to address the feasibility of a portable fluorescent reader with estimated costing of our prototype device. Please see the Amendments to the Manuscript in Response to Reviewer 2 Comment 2b.

Reviewer 3 Comment 2:

'Minor issues:

In lines 55-56, the authors not spin-enhanced FND LFTs achieve a fundamental LOD of 8.2×10^{-19} M, 94,000-fold more sensitive than conventional gold nanoparticles but in practice it is limited by nonspecific binding to NC. If known, it would be helpful for the authors to provide the sensitivity enhancement in practice.'

Response: We have amended the manuscript to state the achievable improvement of FNDs in a real assay system, demonstrated in Miller et al for detection of HIV-1 RNA whereby the factor-fold improvement decreased from 94,000-fold in the model assay to 7,500-fold in the real assay system. The results reported in this study directly comparing the FND assay to the gold nanoparticle assay found a 1,100-fold improvement in the LoD. This reduction from the HIV-1 RNA assay is likely due to higher non-specific binding in a protein detection sandwich

format assay which requires a capture antibody. In contrast, the molecular-based assay detected biotin-modified amplicons that bind to a streptavidin test line.

Amendment to manuscript:

Main Text (Lines 55-60): ‘Our recent work demonstrated the potential of spin-enhanced FNDs in a rapid LFT, achieving a fundamental detection limit of 8.2×10^{-19} M, 94,000-fold more sensitive than conventional gold nanoparticles¹⁵. In a diagnostic assay, however, this limit is not reached because of nonspecific binding of nanoparticles to the membrane, reflected by the 7,500-fold sensitivity improvement achieved for the HIV-1 RNA model assay.’

Reviewer 3 Comment 3. ‘In line 60-79, it is unclear as to why sensitivity has been a challenge with FND biosensors. The previous paragraph highlighted the high-sensitivity of this approach – why did previous research find relatively modest sensitivity (e.g 1.94 ng/mL for SARS-CoV-2 by Hsiao et al.?)’

Response: While FND readout offers advantages in sensitivity, LFT performance is also impacted by assay design, binding kinetics and non-specific binding. This work emphasises selection of high affinity antibody pairs and running buffer optimisation to mitigate the non-specific interactions. We therefore believe, the moderate sensitivity in achieved by Hsiao *et al.*¹⁷ can be explained by utilising the competitive bind assay format, which is less sensitive at low concentrations due to the multivalency of large nanoparticles, challenging to optimise and better suited to small molecule detection and cases where very high analyte concentrations are problematic (hook effect)¹⁸. Furthermore, their assay design utilised smaller (but higher density) FNDs (100nm, ~65x less bright per particle). They used a non-covalent antibody functionalisation method which may lead to dissociation of antibodies from the particles, and low surface densities (which is proportional to binding rate). There are limitations in their optical system: lower excitation power, independent optical paths, a lower magnification objective, and a photomultiplier instead of a camera, all of which could contribute to less efficient optical collection and assay signal-to-noise ratios.

In contrast, our work emphasizes high-affinity antibody pairs, optimized buffer formulations to minimize non-specific binding, larger and brighter (600nm) FNDs utilising stable covalent functionalisation with the hydrophilic polyglycerol coating that’s been shown to reduce non-specific binding by ~4-fold (Miller *et al.*¹⁹, Extended Data Fig. 3a). Our optical system also benefits from higher excitation power density, a single optical path for excitation and collection that allows for more reliable alignment, a higher magnification objective, and a camera for more reliable signal detection and alignment of the FND test line.

Reviewer 3 Comment 4. ‘600 nm particles are generally large for lateral flow applications. Does this lead to restrictions on the types of compatible nitrocellulose, i.e., faster-flowing, larger-pore variants?’

Response: We have observed adequate flow of the 600nm FNDs on the CN95 (Sartorius) membrane used for this work (see response to Reviewer 2, Comment 3) which is a fast-wicking membrane (65-115 s/40mm). We have also evaluated larger FND particle sizes across the same membrane (CN95). Response Figure 11a (below) shows the SNR from the finalised assay pairs (FND-MM08 + bR001) tested with 700nm, 1000nm, and 1200nm FNDs.

The 700nm particle showed similar performance to the 600nm FNDs used in this work, however a drop in SNR was observed for the larger 1 μ m and 1.2 μ m FNDs indicating flow could be impaired with larger particles. Response Figure 11b of the figure below shows the SNR of 600nm FNDs with in-house manufactured strips using CN95 and FF120HP (Whattman) for comparison of a slower wicking membrane (90-140 s/40mm) that showed comparable SNR. Testing a wider range of membrane pore sizes was out of the scope of this study. Alternative smaller FND sizes (120nm and 200nm) have been shown to be highly sensitive viable option (Miller et al.¹⁹) in the case where flexible membrane choice is desirable.

Response Figure 11: (a) The SNR of different size FNDs tested using the same assay format (FND-Ab-MM08 and secondary antibody bR001) on CN95 polystyrene test line strips, showing a reduced SNR with FND size 1000nm and 1200nm. (b) The SNR of 600nm FNDs (FND-AbMM08 and secondary antibody bR001) tested on in-house printed polystyrene test lines with CN95 membrane compared to slower wicking membrane FF120HP (Whattman)

Reviewer 3 Comment 5. 'In line 165, did the authors proceed with all four commercially antibodies? If so, please make this clear.'

Response: We can confirm that we did proceed with all four antibodies and have updated the manuscript accordingly.

Amendment to manuscript:

Main Text (Lines 167-169):

'We proceeded with all four commercial antibodies, found to have stronger binding affinities (K_D values 0.19 nM to 0.34 nM), and for their demonstration as high performance antibodies in literature'

Reviewer 3 Comment 6. 'Although clinical performance shows that the assay performs exceptionally well, down-selecting antibody pairs based on performance against antigen from a single source can lead to bias. Cate et al (<https://pubs.acs.org/doi/10.1021/acsomega.1c01253>) showed marked difference in antibody performance when looking at nucleocapsid proteins from two different vendors

and for those derived from a clinical pool.'

Response: We thank the Reviewer for raising this point about different suppliers. Our initial studies were performed with a combination of three antigen sources (Sino biological, in-house Peter Cherepanov, and in-house Ciara O'Sullivan) and SNR of recombinant antigens were compared with selected pairs. Testing of two antigen sources (Sino biological and O'Sullivan Lab, URV) on the FND assay using the same antibody pair showed similar performance at the select test concentrations (Response Figure 12, t-test paired p-value 0.7621, T=0.3315, DF=3). We agree that simultaneously screening antibody pairs with multiple antigen sources in future would improve robustness of pair selection in future. These results have been added to Supplementary Information Section 3a.

Response Figure 12: Comparison of two different sources of nucleocapsid protein. Performance with commercially available nucleocapsid protein (Sino Biological Inc.), and nucleocapsid protein developed in-house by O'Sullivan Lab, tested on the FND SARS-CoV-2 antigen assay showing similar performance across the same antibody pair (t-test paired p-value 0.7621, T=0.3315, DF=3).

Amendment to manuscript:

Supplementary Information Section 3a (Lines 82-91):

3) Performance evaluation with recombinant antigen

Performance of the two different antigen sources (Sino biological and O'Sullivan Lab, URV) were compared at select test concentrations with the FND SARS-CoV-2 antigen assay during initial assay development.

Response Figure 12 added to Supplementary Information Section 3a as Supplementary Information Fig. 2

Reviewer 3 Comment 7. 'Can the authors speculate on the reasons the FlowFlex assay showed 13x poorer sensitivity than the devices claim? We agree the comparison with an in-house gold assay is the more important result; however, 13x poorer sensitivity than the device literature is significant.'

Response: The poorer sensitivity of commercially available LFTs observed in practice has been reported in literature across many different commercially available devices and can be

partly attributed to the nature of the quality criteria set by regulatory bodies (FDA, WHO, European Centre for Disease Prevention and Control), which had less stringent standards for approval during the pandemic (sensitivity of 80% for PCR-positive specimens) and limited sample number requirements, limiting accurate evaluation of the reproducibility of the test. As described in Scheiblaue *et al.*²⁰, nearly all of the 122 CE-marked tests evaluated in this study had claimed IFU sensitivity values >90% for PCR positive samples, contrary to their findings, which may be attributed to preselection of high PCR positive samples and/or few samples tested.

With regards to this particular study evaluating performance with the inactivated virus sample, discrepancies between results may also be attributed to the lack of standardised samples. For instance, our study used gamma-irradiated whole virus and the FlowFlex Rapid Antigen test evaluation reported using heat-inactivated whole virus sample for which some reports suggest the method of inactivation may impact device performance, as well as a lack of well-defined VTM formulations used²¹.

Reviewer 3 Comment 8. ‘Given that infectious virus determined by culture corresponds to a level of roughly 10^6 copies/mL, is there concern this highly sensitive test will continue to remain positive after the infectious period similar to RNA tests?’

Response: We thank the Reviewer for the comment and believe this is an important point to address. The human challenge trial data found extended RNA positivity in after day 14 in 15/18 patients, however at low viral load concentrations generally less than 10^3 copies/mL (Killingley *et al.*², figure 2a). Our limit of detection at 5.6×10^4 copies/mL has good sensitivity to detect most infectious patients, while not overly sensitive to the low levels of residual circulating RNA that have been observed in some patients. This is described in Lines 344-351 of the main text:

‘(ii) RT-qPCR can suffer from extended RNA positivity where thresholds for infectiousness in relation to Ct values has yet to be fully established⁵². Notably, Killingley *et al.*²⁵, reported quantifiable virus by RT-qPCR was still present by day 14 necessitating extended quarantine. In contrast, viable virus by cell culture, often used as a better indicator of infectiousness, showed viral clearance by day 10.2 on average and no later than day 12. This supports compounding evidence that Ag-LFTs are good determinants of infectiousness, and more closely aligns with the FND assay thresholds where our fitting shows latest detection at day 12 (Fig 4a).’

Reviewer 3 Comment 9. ‘Both antibodies selected by the authors have been used in various publications regarding LFTs – the authors may choose to cite a couple of these to highlight that these are generally considered high-quality antibodies for this target.’

Response: We thank the Reviewer and have cited the relevant literature.

Amendments to manuscript :

Main Text (Lines 167-172):

‘We proceeded with all four commercial antibodies, found to have stronger binding affinities (K_D values 0.19 nM to 0.34 nM), and for their demonstration as high performance antibodies

in literature. This included AbMM08, AbR004, and AbR001 ranking amongst the top 5 performing antibody pairs in a large evaluation of 1021 SARS-CoV-2 antibody pairs (Cate et al.³⁷) and numerous early reports establishing their high quality during the pandemic's initial test development phases³⁸⁻⁴⁰.

Reviewer 3 Comment 10. 'In the methods, line 391-393, please state the concentration of particles in addition to the volume which is already provided. This will make the results more reproducible for other investigators.'

Response: We thank the Reviewer and have updated the manuscript to state the concentration of particles.

Amendments to manuscript:

Methods (Lines 428-431):

'Antibody functionalisation concentration was adapted for 600nm particles considering surface area to volume ratio where 2.71µg of antibody (1mg/mL) was added to 100µL of particles at 1 mg/mL stock concentration.'

Reviewer 3 Comment 11. 'In lines 413-418 the sandwich format is described. The authors state the sample, FNDs and bAb01 are added and allowed to bind for 10 minutes. Is the LFT then placed in and allowed to run for 15 minutes similar to the direct bind format? As written, it is unclear what happens after the 10 minute incubation.'

Response: We thank the Reviewer for spotting this error and have reworked the text to make our protocol as clear as possible.

Amendments to manuscript:

Methods (Lines 450-462): Reorganised methods description
FND lateral flow testing

'The lateral flow assays were run using commercially available strips with polystreptavidin printed test lines (Global Access Diagnostics). FND-PG-AbMM08 was diluted in PBS to concentration of 26.4fM. For the direct bind format, 5µL of the FNDs + 50µL of biotinylated nucleocapsid protein diluted in running buffer (150mM Tri-HCL pH 8, 2% IGEPAL CA-630, 100mM NaCl, 0.05%Tween20, 0.8%Casein) was added to wells in a 96-well plate and allowed to bind for 10mins. The strips were added to each well and allowed to run for ~15 mins. Strips were allowed to dry before read-out described in the following section. Full microscope set up is described in Miller et al¹⁵. For the sandwich format assay, FND-PG-AbMM08 was diluted in PBS to concentration of 26.4 fM and secondary biotinylated AbR001 was diluted in PBS to 513nM. 5µL of the FNDs + 49µL of sample (nucleocapsid protein or gamma-irradiated virus) diluted in running buffer + 1µL of bAb01 (513nM) was added to wells in a 96-well plate and allowed to bind for 10mins. Strip were added to the wells and run for 15 mins, followed by test line read-out of dry strips.'

Reviewer 3 Comment 12. 'Please include information as to how the samples were run in the

“evaluation of clinical nasal swab samples”. Specifically, were both antibodies added to the well with a sample, allowed to bind for 10 minutes, and then run up the LFT? Was the LFA allowed to dry prior to imaging?’

Response: We have updated the methods to clarify the protocol followed for lateral flow testing of clinical swab samples (see methods section ‘Evaluation of clinical swab samples’)

Amendments to manuscript:

Methods (Lines 508-512):

‘For lateral flow testing, 40µL of swab sample in VTM was added to the well and mixed with 9µL of 5X running buffer (see Supplementary Information section 1b for buffer formulation) + 5µL of the FNDs (26 fM) + 1µL of AbR001-b (513nM) and allowed to bind for 10mins. Strips were added to wells and ran for 15mins. The strips were allowed to dry before read-out described in the previous section.’

Reviewer 3 Comment 13. ‘In supplementary information table 1 please indicate what the range in brackets refers to — if it is 95% Cis please note this.’

Response: Please see edits to table above addressing Reviewer #1 comment 5

Reviewer 3 Comment 14. ‘Overall, the paper presents an exciting clinical demonstration highlighting the utility of FND based LFTs. We recommend accepting the paper pending minor edits.’

Response: We thank the Reviewer for this excellent endorsement and recommendation to publish.

Reviewer #4 (Remarks to the Author):

Reviewer #5 (Remarks to the Author):

I co-reviewed this manuscript with one of the reviewers who provided the listed reports. This is part of the Nature Communications initiative to facilitate training in peer review and to provide appropriate recognition for Early Career Researchers who co-review manuscript

Response Reference List

1. Killingley, B. *et al.* Safety, tolerability and viral kinetics during SARS-CoV-2 human challenge in young adults. *Nat Med* **28**, 1031–1041 (2022).
2. Glenn, P. *et al.* Comparison between Nasal and Nasopharyngeal Swabs for SARS-CoV-2 Rapid Antigen Detection in an Asymptomatic Population, and Direct Confirmation by RT-PCR from the Residual Buffer. *Microbiol Spectr* **10**, e02455-21 (2022).
3. Brümmer, L. E. *et al.* Accuracy of Novel Antigen Rapid Diagnostics for SARS-CoV-2: A Living Systematic Review and Meta-Analysis. *PLoS Medicine* vol. 18 (2021).
4. Terada, D., Sotoma, S., Harada, Y., Igarashi, R. & Shirakawa, M. One-Pot Synthesis of Highly Dispersible Fluorescent Nanodiamonds for Bioconjugation. *Bioconjug Chem* **29**, 2786–2792 (2018).
5. Lateral Flow Immunoassays - Jackson ImmunoResearch. <https://www.jacksonimmuno.com/technical/products/applications/elisa/lateral-flow/immunoassays-introduction>.
6. Protocol for Lateral Flow Immunoassay - Creative Diagnostics. <https://www.creative-diagnostics.com/lateral-flow-immunoassay-protocol.htm>.
7. Grant, B. D. *et al.* A SARS-CoV-2 coronavirus nucleocapsid protein antigen-detecting lateral flow assay. *PLoS One* **16**, e0258819 (2021).
8. Cherkaoui, D. *et al.* CRISPR-assisted test for *Schistosoma haematobium*. *Sci Rep* **13**, 4990 (2023).
9. Gasperino, D., Baughman, T., Hsieh, H. V., Bell, D. & Weigl, B. H. Improving Lateral Flow Assay Performance Using Computational Modeling. *Annual Review of Analytical Chemistry* **11**, 219–244 (2018).
10. Chiu, May L *et al.* Matrix Effects—A Challenge toward Automation of Molecular Analysis. *JALA: Journal of the Association for Laboratory Automation* **15**, 233–242 (2010).
11. Frutiger, A. *et al.* Nonspecific Binding—Fundamental Concepts and Consequences for Biosensing Applications. *Chem Rev* **121**, 8095–8160 (2021).
12. Bachman, C. M. *et al.* Clinical validation of an open-access SARS-COV-2 antigen detection lateral flow assay, compared to commercially available assays. *PLoS One* **16**, e0256352 (2021).
13. Feuerstein, G. Z. *et al.* The Use of Near-Infrared Light-Emitting Fluorescent Nanodiamond Particles to Detect Ebola Virus Glycoprotein: Technology Development and Proof of Principle. *Int J Nanomedicine* **15**, 7583–7599 (2020).
14. Wang, C. *et al.* Ultrasensitive and Simultaneous Detection of Two Specific SARS-CoV-2 Antigens in Human Specimens Using Direct/Enrichment Dual-Mode Fluorescence Lateral Flow Immunoassay. *ACS Appl Mater Interfaces* **13**, 40342–40353 (2021).
15. Land, K. J., Boeras, D. I., Chen, X. S., Ramsay, A. R. & Peeling, R. W. REASSURED diagnostics to inform disease control strategies, strengthen health systems and improve patient outcomes. *Nat Microbiol* **4**, 46–54 (2019).
16. IPP Secretariat. Third Implementation Report – IPPS. <https://ippsecretariat.org/publication/third-implementation-report/> (2024).
17. Wei-Wen Hsiao, W. *et al.* Fluorescent nanodiamond-based spin-enhanced lateral flow immunoassay for detection of SARS-CoV-2 nucleocapsid protein and spike protein from different variants. *Anal Chim Acta* **1230**, 340389 (2022).

18. Pedreira, J. *et al.* A Comprehensive Review of Competitive Lateral Flow Assays Over the Past Decade. *Lab Chip* (2025) doi:10.1039/D4LC01075B.
19. Miller, B. S. *et al.* Spin-enhanced nanodiamond biosensing for ultrasensitive diagnostics. *Nature* **587**, 588–593 (2020).
20. Scheiblauer, H. *et al.* Comparative sensitivity evaluation for 122 CE-marked rapid diagnostic tests for SARS-CoV-2 antigen, Germany, September 2020 to April 2021. *Eurosurveillance* **26**, (2021).
21. Sydney, S. *et al.* Limit of Detection for Rapid Antigen Testing of the SARS-CoV-2 Omicron and Delta Variants of Concern Using Live-Virus Culture. *J Clin Microbiol* **60**, e00140-22 (2022).

Response to Reviewers

Manuscript ID: NCOMMS-24-55657-B

Title: Quantum-enhanced nanodiamond rapid test advances early SARS-CoV-2 antigen detection in clinical diagnostics

Authors: Alyssa Thomas DeCruz, Benjamin S. Miller, Da Huang, Max McRobbie, Felix Donaldson, Laura E. McCoy, Ciara K. O'Sullivan, Johannes C. Botha, Eleni Nastouli, Rachel A. McKendry

We thank the reviewers for their feedback and recommendations. Below we have systematically responded to the Reviewers comments.

Reviewer #1 (Remarks to the Author):

The authors have replied to all our comments and kindly considered all our suggestions. We are very satisfied with the current version of the manuscript and we have no additional comments. Therefore, we can recommend its acceptance in the journal.

Response to Reviewer #1:

We thank the Reviewer for their recommendation to publish.

Reviewer #2 (Remarks to the Author):

The authors identify the core innovation of this work as the translation of a previously proposed spin-enhanced LFT model into clinical applications. They conducted a series of experiments to evaluate its practical usability and reported encouraging results, which contribute to advancing the commercialization or industrial application of LFT technology. However, whether the application of an existing technical model to clinical settings meets the innovation threshold expected by Nature Communications is not straightforward to assess from my disciplinary perspective. The editorial team, being more familiar with the journal's criteria for novelty, is in a better position to make this judgment. With regard to the authors' response, the following points may benefit from further clarification or revision:

Reviewer #2, comment 1:

1. If the authors consider the clinical translation of the technique to be the primary innovation, the current abstract may not fully reflect this emphasis. While it provides a detailed account of clinical results and their potential implications, it gives limited explanation of how the spin-enhanced LFT ensures specificity and sensitivity in a clinical context. Briefly outlining the key methodological principles would improve clarity.

Response to Reviewer #2, comment 1:

We thank the reviewer for their feedback on the abstract. We have revised the abstract to improve the clarity of the principles underlying this work.

Amendments to Manuscript:

Abstract:

“Quantum biosensors, which harness quantum effects to detect biomarkers, could address the urgent need for more sensitive rapid diagnostics. Lateral flow tests using nitrogen-vacancy centres in nanodiamond labels offer high sensitivity and robustness by controlling the spin-dependent fluorescence to remove background. This is particularly important in complex and variable clinical samples. However, to date only model systems have been studied with few clinical samples. Here we show results of a clinical evaluation of a spin-enhanced nanodiamond test for SARS-CoV-2 antigen with 103 upper respiratory tract swab samples. We find 95.1% sensitivity ($Ct \leq 30$) and 100% specificity benchmarked against RT-qPCR, with no cross-reactivity to influenza A, RSV, and Rhinovirus. Modelling with patient data yields a mean of 2.0-days earlier detection compared to conventional gold-nanoparticle tests (just 0.6 days after RT-qPCR) with 2.2-fold more patients detected on the first day of symptom onset, potentially reducing the transmission risk and protecting populations.”

Reviewer #2, comment 2:

2. The table listing general cost for detection methods appears not to be directly derived from the current study. Including such a table may not be appropriate unless the data specifically reflect the detection work described in this manuscript. A cost analysis focused on the method proposed in this work would be more relevant and informative.

Response to Reviewer #2, comment 2:

We thank the reviewer for their feedback on the reader costing table and note that this portable reader system was not used in this work. We have included the proposed reader costing table on the development of a portable smartphone reader to support the claim that this assay could be made suitable for the point-of-care. This is an important consideration and sets the agenda for future work. We have updated the accompanying text in the Supplementary Information to make this clearer: “Although the measurements in this paper were performed using a fluorescence microscope, they could in principle be performed with a low-cost device.” and have added “proposed portable fluorescent reader” to make it clear this is hypothetical.

Reviewer #3 (Remarks to the Author):

The authors answered all reviewer questions and comments satisfactorily. Great work!

Response to Reviewer #3:

We thank the Reviewer for their positive endorsement of our paper.

Reviewer #4 (Remarks to the Author):

Reviewer #5 (Remarks to the Author):
